# Early mucosal responses following a randomised controlled human inhaled infection with attenuated *Mycobacterium bovis* BCG

The development of an effective vaccine against *Mycobacterium tuberculosis* is hampered by an incomplete understanding of immunoprotective mechanisms. We utilise an aerosol human challenge model using attenuated *Mycobacterium bovis* BCG, in BCG-naïve UK adults. The primary endpoint of this study (NCT03912207) was to characterise the early immune responses induced by aerosol BCG infection, the secondary endpoint was to identify immune markers associated with in-vitro protection. Blinded volunteers were randomised to inhale $1 \times 10^7$ CFU aerosolised BCG or 0.9% saline (20:6); and sequentially allocated to bronchoscopy at day 2 or 7 post-inhalation (10 BCG, 3 saline each timepoint). In the bronchoalveolar lavage post-aerosol BCG infection, there was an increase in frequency of eosinophils, neutrophils, NK cells and Donor-Unrestricted T cells at day 7, and the frequency of antigen presenting cells decreased at day 7 compared with day 2. The frequency of interferon-gamma+ BCG-specific CD4+ T cells increased in the BAL and peaked in the blood at day 7 post-BCG infection compared to day 2. BAL cells at day 2 and day 7 upregulated gene pathways related to phagocytosis, MHC-II antigen loading, T cell activation and proliferation. BCG's lack of key virulence factors and its failure to induce granulomas, may mean the observed immune responses do not fully recapitulate *Mycobacterium tuberculosis* infection. However, human infection models can provide unique insights into early immune mechanisms, informing vaccine design for complex pathogens.

Tuberculosis (TB), caused predominantly by *Mycobacterium tuberculosis* (*M.tb*), remains one of the world's biggest killers from a single infectious agent, despite widespread use of the only licenced vaccine, Bacille Calmette–Guérin (BCG)[1]. An effective TB vaccine needs to improve upon natural immunity, so a better understanding of protective immunity is needed[2].

The predictive value of preclinical animal models used in TB vaccine development is unclear[3,4]. Identifying a biomarker which correlates with protective immunity could be used in pre-clinical and clinical studies to facilitate the development and selection of vaccine candidates[5]. Class II-restricted CD4+ T cells and interferon-gamma (IFN-γ) are necessary but insufficient to prevent infection or disease[6–8]. Other possible protective immune markers include the monocyte:lymphocyte ratio, neutrophils, NK cells, Donor Unrestricted T cells (DURTs), polyfunctional CD4+ T cells, T cell activation markers, Th17 cytokines and antibodies[5,7–9]. For a pathogen as complex as *M.tb*, it is unlikely that there will be a single, sufficient immune correlate of protection[10].

✉ e-mail: helen.mcshane@ndm.ox.ac.uk

The airway is the natural route of *M.tb* infection and local mucosal responses differ from systemic responses in TB patients and in animal models[4,11]. Human studies of cells and fluid derived from the airway and alveolar spaces could provide important information about early local immunity[12].

Understanding the innate immune response after mycobacterial exposure, and the impact of this response on the subsequent adaptive response is critical for optimal vaccine development[13]. The role of the innate immune response in protection against TB has traditionally been considered to be restricted to the ability of antigen presenting cells (APCs) to prime T cells[14]. Recently, the concept that innate cells such as NK cells mature into variable effector subclasses, influenced by environmental exposure, as well as recent work in epigenetic imprinting of bone-marrow derived monocytes, suggests that a memory-like phenomenon in innate cells may exist, independent of the adaptive response[15]. This could provide novel targets for vaccine candidates. A major impediment to elucidating these mechanisms is the limitations of field studies to capture early immune events, where defining the dose and the timing of infection is not possible. Controlled Human Infection Models (CHIMs) can be used to evaluate candidate vaccine efficacy but also for defining immune responses after a defined time-point infection[16]. It would be unethical to administer virulent *M.tb*[17], but we have established a mycobacterial CHIM using aerosolised live attenuated *Mycobacterium bovis* (BCG)[18].

In this study we interrogate the early mucosal and systemic immune responses following inhaled aerosolised mycobacterial infection in *M.tb* and BCG-naïve adults.

## Results

### Enrolment

Groups 1-2 were screened, enrolled and followed up between the 19th March 2019 and the 30th July 2020 (Fig. 1). Twenty volunteers inhaled $1 \times 10^7$ CFU aerosolised BCG Danish and 6 volunteers inhaled 0.9% saline; 13

volunteers had a bronchoscopy with BAL collected at day 2 (D2, Group 1) post-inhalation (10 BCG, 3 saline) and 13 at D7 (Group 2). There were no withdrawals, and all volunteers completed the study. Due to the impact of the COVID-19 pandemic, blood or sputum was unable to be collected for some later follow up visits in Group 2 volunteers (from D56), in accordance with UK government policy (Supplementary Table 1A). Baseline characteristics are presented (Supplementary Table 1B).

### Safety

There were no serious adverse events or suspected unexpected serious adverse reactions, no withdrawals due to safety concerns and no pre-defined study stopping or holding rules were activated.

One D2 bronchoscopy was reported as normal but mild mucosal erythema at the carina was noted and one D2 bronchoscopy was reported as abnormal with mild erythema throughout and friable airways. Both volunteers were asymptomatic at the time of bronchoscopy but had reported mild respiratory adverse events (AEs) on D0 and/or D1 post-BCG inhalation. There was no further action recommended by the respiratory consultant for either volunteer. All other bronchoscopies were reported as normal.

The majority of AEs post-BCG were mild, and all spontaneously resolved. AEs were consistent with those reported in our previous study (Supplementary Fig. 1 and Supplementary Note 1)[18].

All volunteers maintained transfer factor for carbon monoxide (TLCO) above 80% predicted post-BCG inhalation. There was no difference in TLCO between BCG and saline volunteers at D7 post-inhalation (median % change from baseline: −5.7% BCG (IQR -13.8;1.3), −5.4% saline (−8.5;0.8), *p* = 0.5 Mann-Whitney).

### BCG was detected in the bronchoalveolar lavage (BAL) from all BCG-infected volunteers

BCG CFU quantified from the reconstituted vials was 0.5 log lower than that stated on the vial (median $1.45 \times 10^6$ CFU/vial, IQR $1.13 \times 10^6;2.04 \times 10^6$), consistent with previous findings[18].

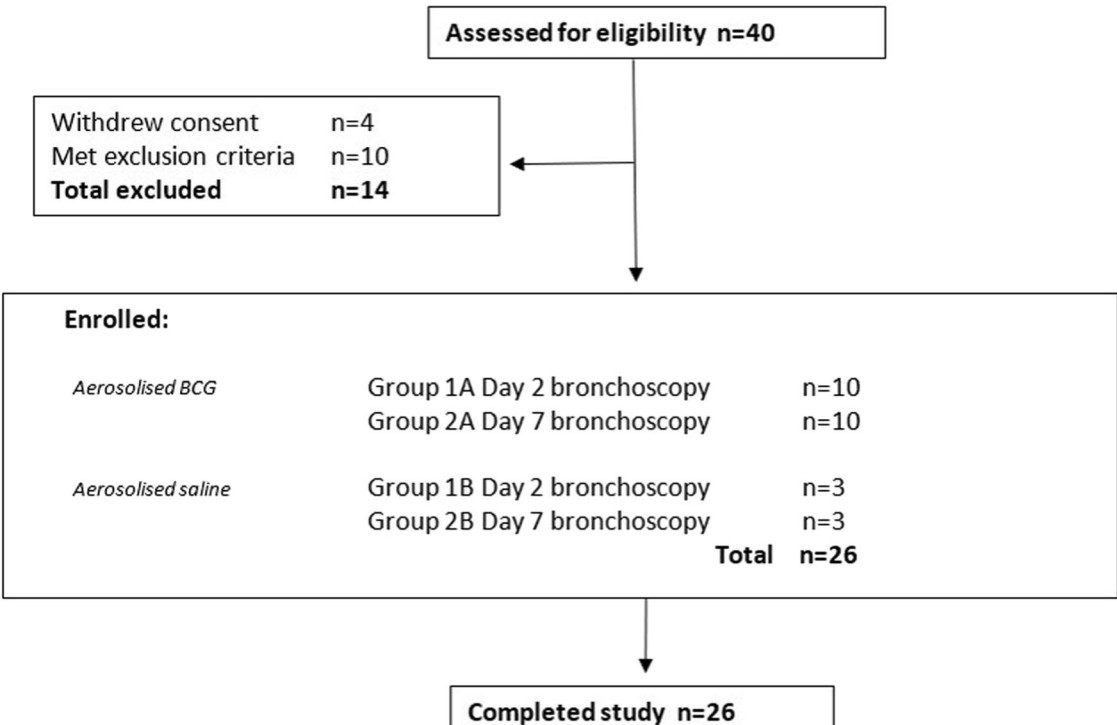

**Fig. 1 | CONSORT diagram of volunteer recruitment.** The first volunteers were screened on 19th March 2019 and enrolment commenced on 29th April 2019. The final volunteer was enrolled on 12th February 2020, and the last volunteer visit was 30th July 2020. Volunteers were randomised to receive either $1 \times 10^7$ CFU aerosol BCG Danish (Group A) or aerosol saline (Group B) with a bronchoscopy at either Day 2 (D2; Group 1) or Day 7 (D7; Group 2) following inhalation. Volunteers were blinded to intervention. Reasons for exclusion during screening included blood dyscrasias, respiratory disease or poor baseline lung function, IGRA positivity.

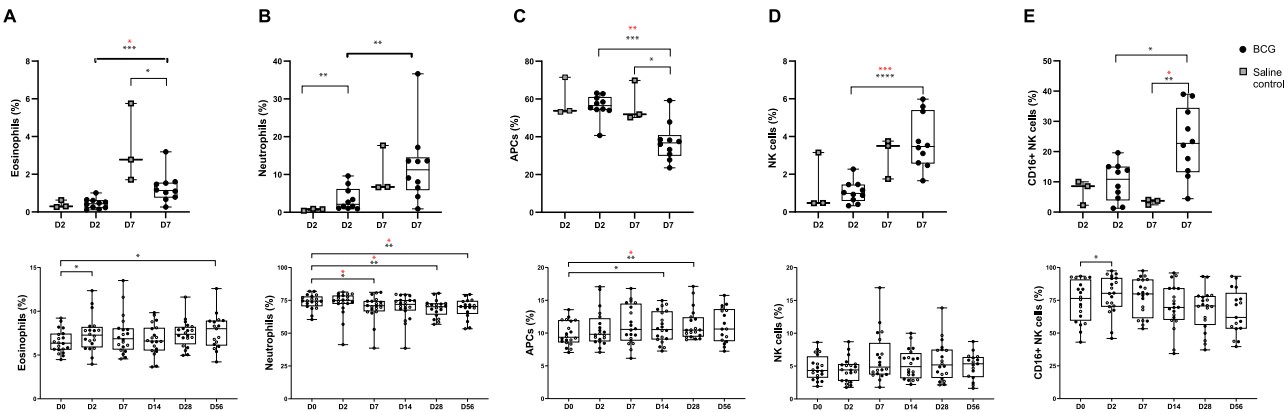

**Fig. 2 | Frequency of innate cells in BAL (upper panels) and whole blood (lower panels) post-BCG infection.** BAL post-BCG or saline inhalation, or blood post-BCG inhalation. Stained with Panel 1. **A** Eosinophils were defined as live singlets, Lin- (CD19, CD56, CD3), CD16- Siglec8+ and shown as % CD3- cells. BAL D7 control v BCG $p = 0.03$, BCG D2 v D7 $p = 0.0007$; after correction BCG D2 v D7 $p = 0.01$. Blood baseline to D2 $p = 0.03$, D56 $p = 0.045$. **B** Neutrophils were defined as live singlets, Lin- (CD19, CD56, CD3), CD16+ , CD66b+ . Shown as % CD3- cells. BAL D2 control v BCG $p = 0.007$, BCG D2 v D7 $p = 0.009$. Blood baseline to D7 $p = 0.03$, D14 $p = 0.07$, D28 $p = 0.003$, D56 $p = 0.003$. After Dunns correction D7 $p = 0.02$, D28 $p = 0.03$, D56 $p = 0.02$. **C** Antigen presenting cells (APCs) refer to macrophages, monocytes and dendritic cells and were defined as live singlets, CD19-CD3-CD56-CD66b-CD45+ and shown as % leucocytes, defined as live singlets, CD19-. BAL D7 control v BCG $p = 0.003$, BCG D2 v D7 $p = 0.0007$; after correction BCG D2 v D7 $p = 0.002$. Blood baseline to D14 $p = 0.03$, D28 $p = 0.006$. After Dunns correction D28 $p = 0.01$. **D** Natural Killer (NK) cells defined as CD19-, CD3-, CD56+ to identify the total NK cell

population as % CD3- cells; or **E** CD16+ NK cells as % total NK cell population. BAL NK cells BCG D2 v D7 $p < 0.0001$, after correction $p = 0.0009$. BAL CD16+ NK cells D7 control v BCG $p = 0.007$, BCG D2 v D7 $p = 0.02$. After correction D7 control v BCG $p = 0.01$. Blood baseline to D2 $p = 0.04$. Differences in cell frequency between BAL groups were calculated using a two-sided Mann-Whitney and corrected with Dunns; and in the blood, differences between baseline samples and corresponding time-points following BCG inhalation were calculated using a paired two-sided Wilcoxon signed rank test against baseline only and corrected with Dunns. Statistically significant differences are presented in the figure, red asterix indicates significant differences after Dunn's correction. *$p < 0.05$, **$p < 0.01$, ***$p < 0.001$, ****$p < 0.0001$. See Supplementary Note 2 and source data for sample sizes per group. Source data are provided as a Source Data file. BAL: Black dots indicate BCG, grey squares saline control. Blood: Solid dots indicate Group 1 (D2 bronchoscopy) volunteers and circles represent Group 2 (D7 bronchoscopy) volunteers. Box and whisker plots show median, IQR and min/max.

Live BCG, using the Mycobacterial Growth Indicator Tube (MGIT) system, was detected in the BAL fluid of 6 (60%) volunteers at D2 post-infection but only 2 (20%) of volunteers at D7 post-infection. BCG was detected, by HAIN™ genotyping, in the BAL fluid from all twenty volunteers post BCG-infection, which could represent live, dead and/or fragmental BCG material. There was no BCG detected in the endobronchial biopsies using a Ziehl-Neelsen stain (Supplementary Fig. 2). There was no BCG detected in the induced sputum samples at 3- or 6-months post BCG-inhalation.

## No difference in Mycobacterial Growth Inhibition Assay (MGIA) using peripheral blood mononuclear cells (PBMCs)
There was no difference in BCG growth using the MGIA with PBMCs post-BCG inhalation compared to baseline (Supplementary Fig. 3).

## Innate cells increased in the airway at Day 7 post-BCG inhalation
There was an increase in frequency of eosinophils, neutrophils and NK cells in the airways of volunteers infected with aerosol BCG at D7 compared with D2 (Fig. 2), though the difference in neutrophils lost significance after correction for multiple comparisons. (Eosinophils: median D2 BCG 0.4% (IQR 0.2;0.6) v D7 BCG 1.1% (0.8;1.6), $p = 0.01$; NK cells: D2 BCG 0.98% (0.6;1.5) v D7 BCG 3.5% (2.6;5.4), $p = 0.0009$, Mann-Whitney with Dunn's correction). The cytotoxic CD16+ NK cell subset increased in the airway at D7 post BCG compared to saline (saline median 3.7% (IQR 2.4;4.1) v D7 BCG 22.7% (13.2;34.4), $p = 0.01$, Mann-Whitney with Dunn's correction). The median neutrophil frequency at D7 in the BAL was 11.3% (IQR 5.9;14.5), indicating an airway neutrophilia (defined as >4.3% total BAL cells)[19] which wasn't present at D2 (2.1% (1.1;6.2)). There was no corresponding significant change in the blood, after correction, except for neutrophils where the frequency reduced from D7 compared to baseline (Fig. 2B; D0 74.4% (70.6;77.8) v D7 70.9% (66.8;74.3) $p = 0.02$; D28 70.1% (64.3;72.5) $p = 0.03$; D56 70.3% (64.85;74.4) $p = 0.02$, Wilcoxon with Dunn's correction).

The frequency of antigen presenting cells in the airways (APCs; includes macrophages, dendritic cells and monocytes) was reduced at D7 post BCG-infection compared with D2 (Fig. 2C; median D2 56.5% (IQR 54.3;61.1) v D7 36.8% (30;40.8), $p = 0.002$, Mann-Whitney with Dunn's), but there was a trend for increased APCs in the airway wall at D7 compared with controls (measured by immunohistochemistry (IHC) staining of endobronchial biopsies; defined as CD68+ cells, Supplementary Fig. 2; controls 0.1% slide (0.03;0.4), D2 0.2% (0.2;0.4), D7 0.5% (0.5;0.5)). The frequency of circulating APCs increased in the blood from D14 post-BCG infection and remained significant at D28 after correction (Fig. 2C; D0 9.4% (8.6;12.0) v D28 10.5% (9.5;12.4), $p = 0.01$, Wilcoxon with Dunn's).

The frequency of CD3+ CD56+ cells (natural killer-like T cells) and γδ T cells increased in the BAL at D7 post-BCG infection compared to D2 (Fig. 3; NKT-like: D2 median 0.2% (IQR 0.2;0.4) v D7 1.4% (0.9;2.4), $p = 0.0006$; γδ: D2 1.0% (0.6;2.1) v D7 6.8% (3.2;13.6), $p = 0.04$). In most BAL samples, there were insufficient cells to stain for mucosal associated invariant T cells (MAITs), CD161+ T cells and invariant natural killer cells (iNKT cells). However there was a trend for an increase in the frequency of iNKTs, CD161 + T cells and MAITs at D7 compared to D2 in the BAL (Fig. 3; iNKT: D2 0.02% (0.01;0.3), D7 0.4% (0.2;0.8); CD161 + : D2 0.5% (0.5;1.2), D7 6.6% (4.3;9.9); MAITs: D2 0.3% (0.3;0.5), D7 2.1% (1.6;3.1). In contrast, the frequency of iNKT cells and γδ T cells fell in the blood post-BCG infection, but significance was lost after correction for multiple comparisons (Fig. 3). iNKT, MAIT and CD161+ T cells were further divided into CD4+ and CD8+ subsets (Fig. 3F–H). For iNKTs and CD161+ T cells there was an increase in the frequency of the CD4+ subsets in the BAL at D7 compared with D2 post-BCG infection (iNKT mean difference 60.1 (95%CI 103.7;16.4), $p = 0.02$; CD161+Tcells 24.7, (48.1;1.3), $p < 0.0001$ and a corresponding reduction in double negative iNKT cells (−52.6 (−11.61;−93.5), $p = 0.02$; mixed effects with Bonferroni correction).

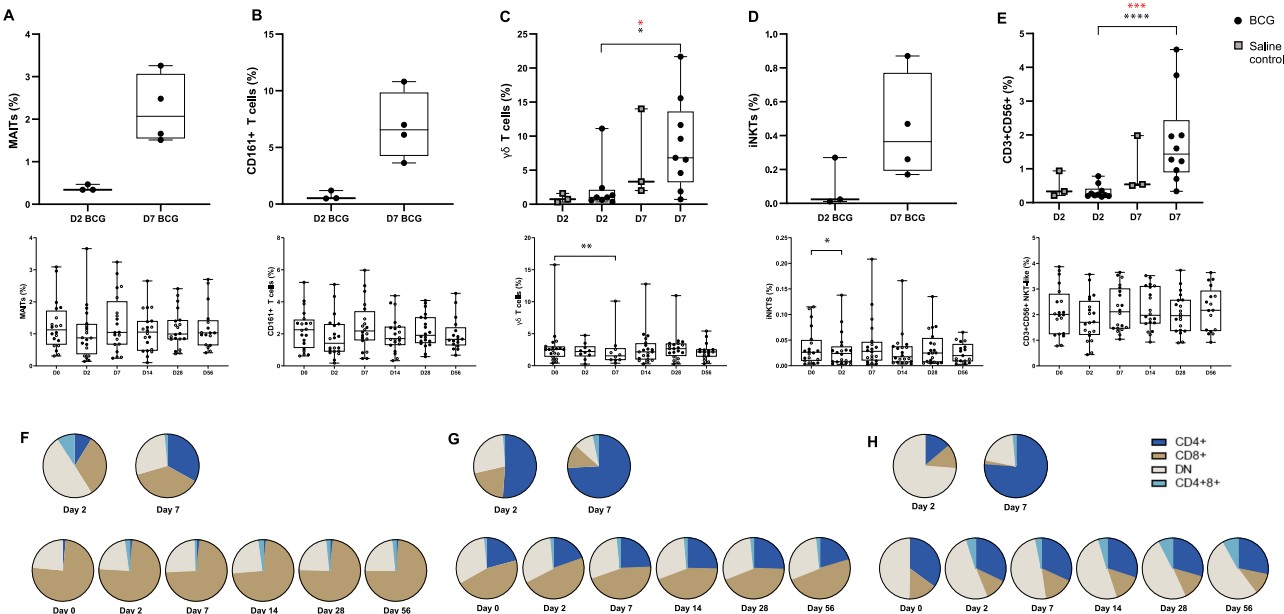

**Fig. 3 | Frequency of Donor Unrestricted T cells (DURTs) in BAL (upper panels) and blood (lower panels) post-BCG infection.** BAL post-BCG or saline inhalation or blood post-BCG inhalation stained with Panel 2 (γδ T cells and CD3+ CD56+ cells, whole blood) and Panel 3 (MAITs, iNKTs, CD161 + T cells, PBMCs). As Panel 3 was third priority, there were only sufficient BAL cells for three D2 BCG-infection and four D7 BCG-infection BAL samples to be stained for detection of MAIT, iNKT and CD161+ T cells. **A** MAITs defined as live singlets, CD19-, CD14-, CD3+ 5-OP-RU MR1 tetramer + , 6-FP control MR1 tetramer- (NIH Tetramer Core facility); and shown as % CD14-CD19- cells. **B** CD161+ T cells defined as live singlets, CD14- CD19- CD3 + CD161+ and shown as % CD14-CD19- cells. **C** γδ T cells were live singlets, CD19- CD14- CD3+ γδpan+ cells combined with live singlets, CD19- CD14- CD3 + γδ2+ cells. Expressed as % CD14- CD19- cells. BAL: BCG D2 v D7 p = 0.02; after correction p = 0.04. Blood: baseline to D7 p = 0.002. **D** iNKTs defined as live singlets, CD19- CD14- CD3+ , Vα24+ Vβ11+ CD56+ CD3+ cells and shown as % CD14-CD19- cells. Blood: baseline to D2 p = 0.03. **E** NKT-like CD3+ CD56+ cells defined as live singlets, CD19- CD14- CD3+ CD56+ cells and shown as % CD14- CD19- cells. BAL: BCG D2 v D7 p = 0.0001; after correction BCG D2 v D7 p = 0.0006. MAITs (**F**), CD161 + T cells (**G**) and iNKT cells (**H**) were further defined by CD4 CD8 subsets; median % parent. DN: double negative (CD4-CD8-) (**F**–**H** BAL on upper panels and blood on lower panels). Differences in cell frequency between BAL groups were calculated using a two-sided Mann-Whitney and corrected with Dunn's; and in the blood, differences between baseline samples and corresponding timepoints following BCG inhalation were calculated using a two-sided paired Wilcoxon signed rank test against baseline only and corrected with Dunn's. Statistically significant differences are presented in the figure, red asterix indicates significant differences after Dunn's correction. *p < 0.05, **p < 0.01, ***p < 0.001, ****p < 0.0001. See Supplementary Note 2 and source data for sample sizes per group. Source data are provided as a Source Data file. BAL: Black dots indicate BCG, grey squares saline control. Blood: Solid dots indicate Group 1 (D2 bronchoscopy) volunteers and circles represent Group 2 (D7 bronchoscopy) volunteers. Box and whisker plots show median, IQR and min/max.

There was no difference in the frequency of these cells in the blood.

## T cells increased at Day 7 in BAL after aerosol BCG infection
The frequency of total T cells increased in the BAL at D7 post-BCG infection compared to D2 (D2 median 8.8% (IQR 4.6;10.8) v D7 46.6% (37.2;55.8), p = 0.0003, Mann-Whitney with Dunn's correction) and fell in the blood at D2 post-BCG infection, compared to baseline (D0 19.1% (16.0;21.1) v D2 16.8% (8.9;20.1), p = 0.03, Wilcoxon with Dunn's) (Fig. 4Ai). All CD4+ and CD8+ T cell subsets, as a proportion of total leucocytes, increased in the BAL at D7 post-BCG infection compared with D2 or saline controls (Fig. 4A), with CD4 + T cells being the dominant subset (CD4+: D2 BCG 5.3% (3.8;9.2) v D7 BCG 15% (13.6;33.3), p = 0.03; CD8+: D7 control 1.0% (1.0;4.5) v D7 BCG 5.8% (4.7;9.4), p = 0.03; CD4+ CD8+: D2 BCG 0.1% (0.1;0.4), D7 control 0.08% (0.1;0.2), D7 BCG 2.2% (0.38;8.0), BCG D2 v D7 p = 0.02, D7 control v D7 BCG p = 0.01; Mann-Whitney with Dunn's correction). There was no significant change in the frequency of these subsets in the blood. A CD4+ CD8+ T cell subset was detected in the majority of volunteers in the BAL at D7 post BCG infection (Fig. 4A) but this population was not detected in the blood. There was a trend for the frequency of total T cells and CD4+ T cells to increase at D7 compared with the saline control group on the endobronchial biopsy samples but this did not reach significance (Supplementary Fig. 2; CD3+: control median 1.7% slide (1.0;2.4), D2 1.7% (1.1;2.5), D7 3.1% (2.4;5.7); CD4+: control 0.3% (0.2;0.6), D2 1.0% (0.2;3.1), D7 2.3% (0.9;2.5)).

## Antigen-specific responses increased in the BAL and blood at Day 7 post-aerosol BCG infection
The frequency of IFN-γ+ BCG-specific total T cells, CD4+ T cells and CD4+ CD8+ T cells increased in the BAL at D7 post-BCG infection compared to D2 (Fig. 4B), though significance was lost on correction for the double-positive cells (total T cells: D2 median 0% (IQR 0;0) v D7 24% (6.2;29.2), p = 0.03; CD4+: D2 0% (0;0) v D7 35.1% (11.4;40.5), p = 0.03, Mann-Whitney with Dunn's correction). It was only possible to stain for polyfunctional T cells on two BAL samples (both D7 post-BCG). Double positive IFN-γ+ IL-2+ CD4+ T cells (12.9% or 4.7% CD4+ T cells) or IFN-γ+ TNF-α+ CD4+ T cells (5.7% or 12.1% CD4+ T cells) and triple positive IFN-γ+ IL-2+ TNF-α+ CD4+ T cells (9.2% or 10.2% CD4+ T cells) were detected in these samples.

The frequency of circulating BCG-specific IFN-γ+ or IL2+ CD4+ or CD8+ T cells increased post-BCG infection, but significance was lost on correction (Fig. 4Ci, iii). BCG-reactive TNF-α+ monocytes increased in the circulation from D7 but significance was also lost on correction (Fig. 4D). There was no significant change in the frequency of circulating antigen-specific polyfunctional T cells.

Ex-vivo PPD-specific IFN-γ ELISpot responses significantly increased D7-14 post-BCG infection, peaked at D7 and remained significantly higher than saline controls until D56 (Fig. 4E; D0 BCG median 93 spot-forming cells (IQR 52;203), D7 BCG 803(512;1632), D0 v D7 p = 0.0008; D14 BCG 270 (171;599), D0 v D14 p = 0.003; D7 control 94 (57;186), D7 control v D7 BCG p = 0.001; D56 control 79 (37;113) v D56 BCG 192 (118;282), p = 0.04, Wilcoxon with Dunn's correction).

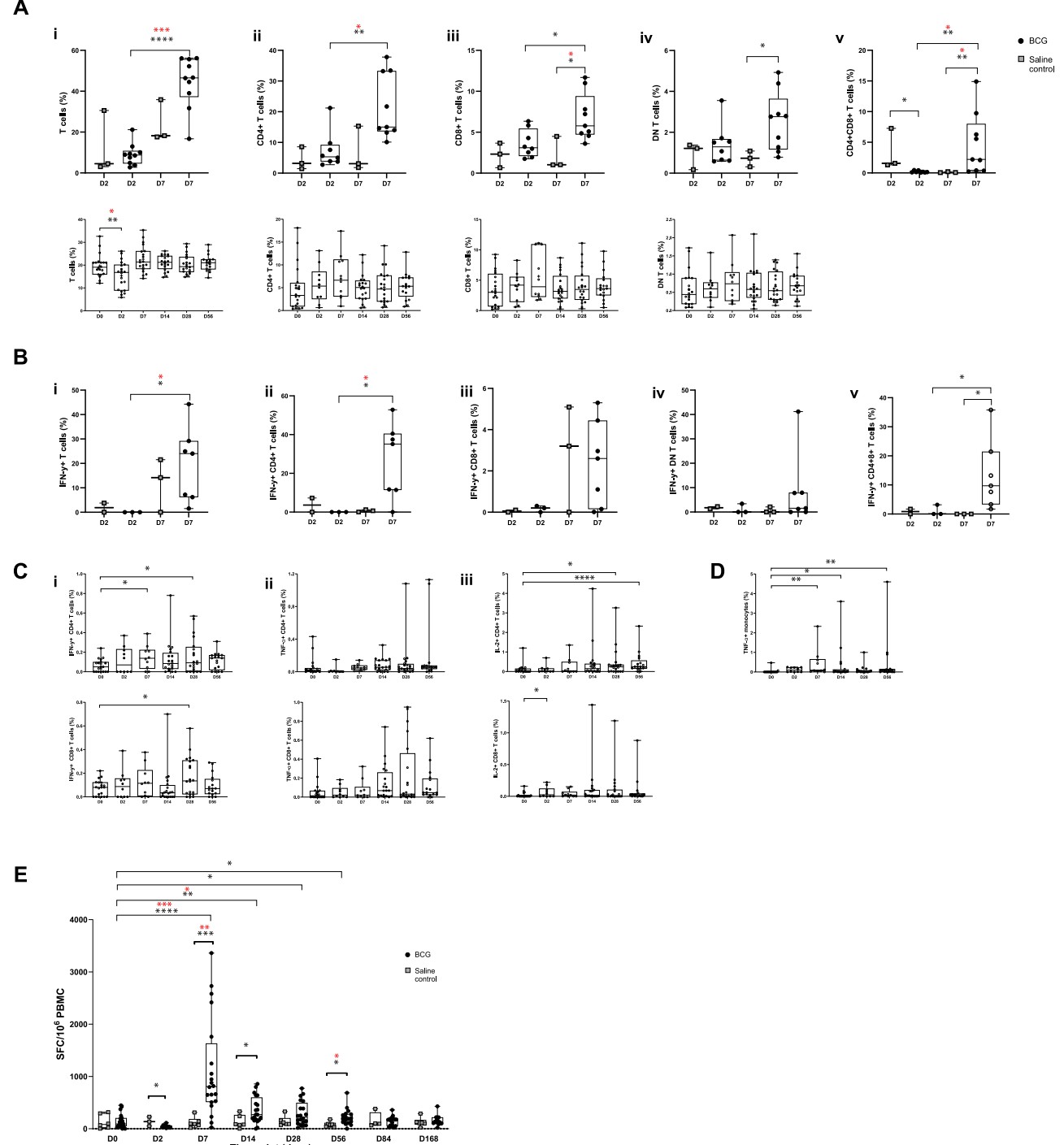

## Antigen-specific humoral responses increase in the serum but not BAL

There was a significant increase in serum PPD-specific IgG antibodies at D56 post-aerosol BCG infection, compared to baseline (D0 median 202 AU (IQR 66;324) v D56 236 (67;306), p = 0.0155, Wilcoxon), but this effect was lost with correction. There was no change in circulating anti-PPD IgA, nor in BAL IgG or IgA antibodies (Fig. 5).

## Tissue resident and activated innate and adaptive cells identified in the BAL by single cell RNA-sequencing

BAL cells from three BCG-infected and three control volunteers from D2 and D7 post-aerosol inhalation were processed for single cell RNA sequencing. Twenty-five cell types were assigned using

canonical markers for cluster gene expression (Fig. 6A). Supplementary Fig. 4 shows the gene expression strategy used for cell annotation. Macrophages were subdivided into non-resident monocytes, activated macrophages and resident macrophage populations. Other cell populations identified in the BAL included neutrophils, NK cells, pDCs and cDCs (cDC1, cDC2 and migratory DCs), γδ2 T cells, classical T cells and MAIT cells (Fig. 6A and Supplementary Fig. 4).

Gene enrichment pathway analysis of alveolar macrophages and dendritic cells showed increased cytokine expression, IFN-γ signalling, leucocyte activation, phagocytosis, migration, apoptosis, antigen processing and presentation pathways either at D2 or D7 post-BCG infection (Fig. 6B, Supplementary Data 1).

**Fig. 4 | T cells and antigen-specific responses in the BAL and blood post-BCG infection. A** BAL (upper panels) and whole blood (lower panels) cells (baseline, D2, D7, D14, D28, D56), Panel 2 staining. T cells (live singlets, CD19-, CD14-, CD3+ ); % Leucocytes (singlets, CD19-). (i) Total T cells; (ii) CD4 + T cells; (iii) CD8+ T cells; (iv) CD4-CD8- T cells (double negative, DN); (v) CD4+CD8+ T cells (double positive, DP). (i) Total T cells: BAL BCG D2 v D7 $p < 0.0001$, after correction $p = 0.0003$; Blood baseline to D2 $p = 0.003$, after correction $p = 0.03$. (ii) CD4 + T cells: BAL BCG D2 v D7 $p = 0.0016$, after correction $p = 0.003$. (iii) CD8+ T cells: BAL D7 control v BCG $p = 0.02$, after correction $p = 0.03$; BCG D2 v D7 $p = 0.02$. (iv) DN T cells: BAL D7 control v BCG $p = 0.04$. (iv) CD4+ CD8+ cells: BAL D2 control v BCG $p = 0.01$; D7 control v BCG $p = 0.009$ after correction $p = 0.02$. BCG D2 v D7 $p = 0.002$, after correction $p = 0.01$. **B** BAL, Panel 2 or 4 staining. Antigen (BCG)-specific IFN-γ+ T-cells in (i) Total T cells; (ii) CD4+; (iii) CD8+; (iv) DN; or (v) CD4+CD8+ T cells. Shown as % parent. (i+ii) Total and CD4 + T cells: BCG D2 v D7 $p = 0.02$ with correction $p = 0.03$. (v) CD4+ CD8+ T cells: D7 control v BCG $p = 0.02$, BCG D2 v D7 $p = 0.03$. **C, D** BCG-stimulated whole blood cells. Panel 2 or 4 staining. **C** Antigen (BCG)-specific IFN-γ+ (i), TNF-α+ (ii) or IL-2+ (iii) CD4+ T cells or CD8+ T cells. Shown as % parent.

**C** (i) IFN-γ+ CD4+ T cells: D7 $p = 0.03$, D28 $p = 0.03$. IFN-γ+ CD8+ T cells: D28 $p = 0.03$. (iii) IL2+ CD4+ T cells: D28 $p = 0.03$, D56 $p = 0.0002$. IL2+ CD8+ T cells: D2 $p = 0.03$. **D** TNF-α+ BCG-reactive monocytes. D7 $p = 0.004$, D14 $p = 0.02$, D56 $p = 0.003$. **E** Antigen (PPD)-specific IFN-γ ELISpot responses in fresh PBMCs. SFC = spot-forming cells. Different timepoints after BCG: Wilcoxon paired test, D7 $p < 0.0001$, D14 $p = 0.0003$, D28 $p = 0.01$, D56 $p = 0.04$; after correction, D7 $p = 0.0008$, D14 $p = 0.003$. BCG v control: Mann-Whitney, D2 $p = 0.049$, D7 $p = 0.0006$, D14 $p = 0.04$, D56 $p = 0.01$; after correction, D7 $p = 0.001$, D56 $p = 0.04$. Statistically significant differences are presented in the figure. BAL: Two-sided Mann-Whitney; blood: two-sided paired Wilcoxon against baseline. Red asterix indicates significant differences after correction (Dunns). *$p < 0.05$, **$p < 0.01$, ***$p < 0.001$, ****$p < 0.0001$. See Supplementary Note 2 and source data (provided as a Source Data file) for sample sizes. BAL: Black dots BCG, grey squares saline control. Blood: Solid dots Group 1 (D2 bronchoscopy), circles Group 2 (D7 bronchoscopy). Box and whisker plots show median, IQR and min/max, crosses show mean.

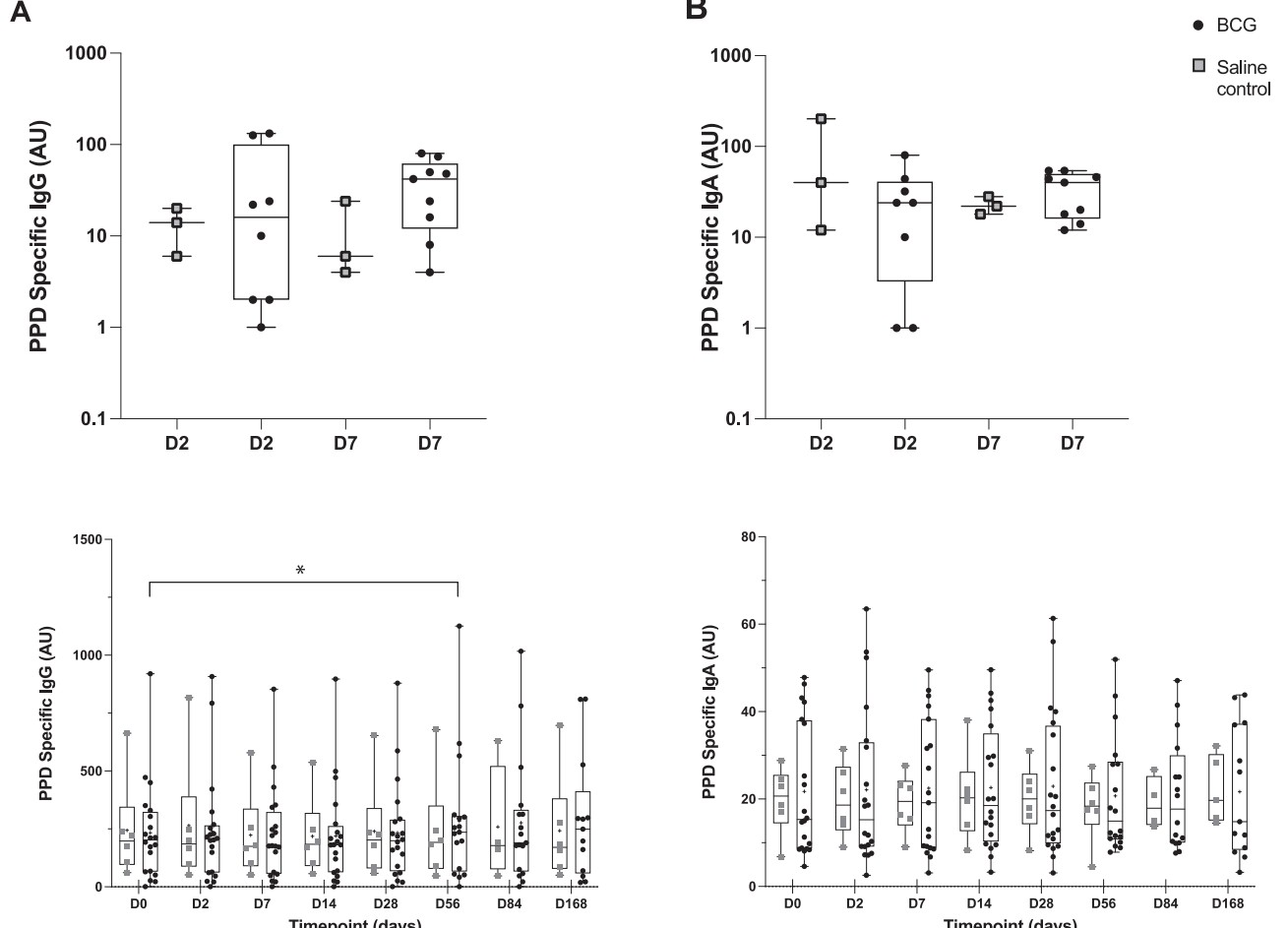

**Fig. 5 | Anti-PPD IgG and IgA antibodies in BAL (upper panel) and serum (lower panel) post-BCG infection.** Total IgG (**A**) and IgA (**B**) against Purified Protein Derivative (PPD) were measured using a standardised in-house indirect ELISA. Arbitrary units (AU). Statistically significant differences are presented in the figure.

IgG serum: D56 $p = 0.0155$ two-sided Wilcoxon, no significance after Dunnett's correction. See Supplementary Note 2 and source data (provided as a Source Data file) for sample sizes per group. Black dots indicate BCG, grey squares saline control. Box and whisker plots show median, IQR and min/max, crosses show mean.

Multiple pathways were upregulated in NK cells at D2 or D7 post-BCG infection, including those involved in cytokine signalling, leucocyte activation, cell cycle, as well as IFN-γ signalling, compared with the saline control group.

Pathway enrichment analysis of DURTs post-BCG infection showed increased expression of genes associated with cellular cytotoxicity, IFN-γ signalling, T cell activation and migration.

Gene pathway enrichment analysis for CD4+ T cells showed an increased expression of genes associated with cytokine signalling, T cell activation and proliferation, IFN responses, activation of nuclear factor-κB (NF-κB) pathway, as well as regulation of migration and apoptosis (Fig. 6B, Supplementary Data 1).

Previous studies have identified several T1-T17 populations in *Mtb* granuloma whose abundance was negatively associated with bacterial

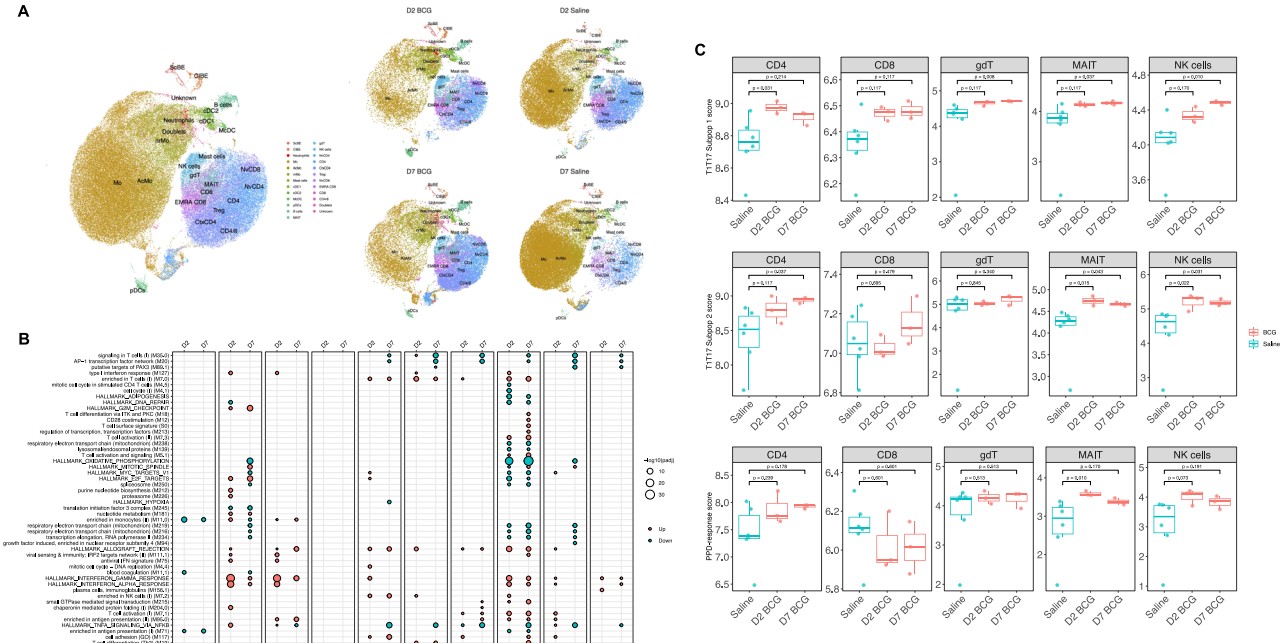

**Fig. 6 | Characterisation of BAL cell at D2 or D7 post-aerosol inhalation using single cell RNA sequencing. A** UMAP (Uniform manifold approximation and projection) plot showing the identity of each cell cluster. Distribution of cells by group. NK cell, γδ T cell, plasmacytoid DC (pDC), conventional DC (cDC), Migratory DC (McDc), Cilliated bronchial epithelial cell (CIBE), Secretory bronchial epithelial cell (ScBE), T regulatory cell (Treg), Proliferating cells (Prl), unidentified cell type (Unk), terminally differentiated effector memory cells re-expressing CD45RA-like CD8 + T cell (EMRA CD8), Naïve (Nv), Cytotoxic-like (Ctx), non-resident Macrophage (nrMo), Macrophage (Mo), Activated macrophage (AcMo). Source data are provided as a Source Data file. **B** Enriched blood transcriptomic modules (BTMs; red, enrichment of upregulated genes; blue, enrichment of downregulated genes) of differentially expressed genes (DEGs) were shown for each cell type on D2 and D7. The differential gene expression analysis was between BCG (D2 or D7) and Saline. One-sided (upper tail) hypergeometric test adjusted by Benjamini-Hochberg multiple testing correction was used to test the enrichment of each gene set. Source data are provided as a Source Data file. **C** The T1-T17 subpopulation 1 score, subpopulation 2 score, PPD-response score in T and NK cell populations following aerosol saline or BCG challenge. Two-sided Dunn's test adjusted by Benjamini-Hochberg multiple testing correction was used to compare the score of each cell type at time points post-aerosolised BCG challenge to that in saline controls. Bars show medians with interquartile ranges (IQR). The upper whisker extends to the largest value no further than 1.5 × IQR from the hinge. The lower whisker extends from the hinge to the smallest value at most 1.5 × IQR from the hinge. N = 6, 3, 3 biologically independent samples for Saline, D2 BCG, D7 BCG, respectively. Source data are provided as a Source Data file.

burden[20]. To investigate whether a similar transcriptional profile was induced at early time points following aerosol BCG challenge, we scored T cell and NK cell populations using the gene signature defining the T1-T17 subpopulation 1, characterised by markers of activation and motility, and T1-T17 subpopulation 2, characterised by markers of cytotoxic effector molecules. The score derived from T1-T17 subpopulation 1 showed increased expression in CD4+ T cells, MAIT cells, γδ T cells and NK cells at either D2 or D7. The score derived from subpopulation 2 predominantly showed increased expression in CD4+ T cells, MAIT cells, and NK cells (Fig. 6C). Additionally, scoring T cell and NK cell populations using a PPD-response signature derived from in-vitro PPD stimulation of BAL cells[21] revealed increased expression in MAIT and NK cells on D2, with a trend towards increased expression in CD4+ T cells on D7.

## Discussion

We describe the early immune events at the airway–parenchymal interface following a precisely timed inhaled mycobacterial infection. Our study characterises these early responses to live mycobacterial exposure in *M.tb*- and BCG-naïve human volunteers.

As with our previous study, aerosol BCG inhalation was well tolerated[18]. We saw mild macroscopic asymptomatic changes in the airway in 2 of 10 volunteers D2 following aerosol BCG. We detected live BCG in the BAL fluid in 60% of subjects D2 after aerosol BCG delivery, which was followed by local and systemic antigen-specific IFN-γ production peaking in the circulation at D7 post-infection. This D7 systemic immune response peak was also seen in our previous BCG aerosol infection study and differs from the slower and broader peak seen 1 to 4 weeks post intradermal BCG vaccination[22,23]. The kinetics of the adaptive immune response following BCG aerosol infection are also earlier than antigen-specific systemic responses to aerosol *M.tb* infection in non-human primate (NHP) models[4,24]. In humans, tuberculin skin test conversion, a measure of delayed Type IV hypersensitivity to PPD, appears as early as 2 weeks in TB case contact field studies but with TST conversion peaking 4-8 weeks post-exposure[25,26]. There is a similar kinetic for IGRA conversion[27]. Whether IGRA conversion can occur earlier than this, has not, to our knowledge been described.

None of our volunteers developed a systemic BCG infection and all demonstrated clearance of BCG in induced sputum 3 months after exposure. Overall, the immune response described in this study can be associated with successful airway clearance of BCG, though further work is required to understand which responses are specifically associated with protective immunity.

The prepared dose of $1 \times 10^7$ CFU was chosen based on our previous experience of tolerated prepared or loaded dose[18]. As reported in our previous study, the dose administered into the lungs was likely $1 \times 10^4$–$1.5 \times 10^5$ CFU, 1.5–3 log lower than $1 \times 10^7$ CFU, due to the expected losses during nebulisation and the known variability in BCG within the starting vial[18,24,28].

The MGIA has previously detected an enhanced ability to control mycobacterial growth following ID BCG vaccination in humans, with peak growth control at D56[29]. The inability to detect a change in control using the ex-vivo MGIA, despite clinical control of BCG infection,

may be due to lack of sensitivity using PBMCs after an aerosol infection. Improved control has been shown on a murine lung MGIA[29]. Further work is in progress to optimise a BAL MGIA in humans.

The hallmark of an effective early immune response against a mycobacterial infection is understood to be recognition and phagocytosis of mycobacteria, antigen loading onto MHC-II, and subsequent Th1 CD4 + T cell-help to augment intracellular killing[30]. Until now, these early unprimed airway processes were hypothesised from animal models and observations of exposed household contacts, where precise timing and dose quantification is very difficult[4,14,31,32]. We demonstrated upregulation of gene pathways consistent with these processes, in situ early in an *M.tb* and BCG-naive human infection model.

We saw a reduction in alveolar macrophages (AMs) in the airway which may reflect a macrophage disappearance reaction[33], due to infected AMs moving from the airway and alveoli into the parenchyma, or destruction following BCG infection. In their murine model, Cohen et al. demonstrated that airway resident AMs were the primary *M.tb* infection targets, and that only infected AMs migrated into the interstitium[14]. However, as this phenomenon was dependent on ESX-1, which BCG does not express, it is unclear if the same process is occurring here[34]. BAL may contain cells derived from the airway and the parenchyma, further complicating the picture. APCs increased in the endobronchial wall at D7 but fell in the BAL, this could be because of increased infection of AMs in the alveolar spaces compared with the airway and/or subsequent migration out of the alveolar spaces[33].

Impairment of DC-mediated antigen presentation and delayed tracking to draining lymph nodes, with subsequent attenuation of Th1 CD4 + T effector function, is an immune evasion strategy believed to facilitate progression to latency or active disease, based on animal models[35,36]. Here we show that local immune responses, including CD4 + T cell influx and Th1 effector function, do not appear to be delayed following aerosol BCG infection. Gene pathways consistent with antigen presentation, phagosome and apoptotic pathways are upregulated in the airways as early as D2 after BCG aerosol infection. In addition, we found an independently derived gene signature in T cells from *M.tb* granuloma associated with reduced bacterial burden[20], as well as a PPD-response gene signature from in-vitro PPD-stimulated T cells[21], were induced in several T and NK cell populations following aerosol BCG challenge. To our knowledge, these mechanisms have not previously been demonstrated at such an early time-point following mycobacterial infection in-vivo in humans. As these mechanisms may be associated with the in-vivo control of aerosol BCG infection that we see in this model, these results might support the hypothesis that a vaccine which strongly activates Th1 polarising DCs could induce a protective mucosal T cell response against mycobacterial disease. However further work is required to demonstrate that the effect seen here contributes to protective immunity.

Other innate immune cells such as granulocytes, NK cells and DURTs also increased in the airway and upregulated effector function during the first week post-BCG infection. These cells are capable of early intracellular BCG recognition, which has been shown to be important for early clearance in animal studies[35,37,38].

While lung neutrophilia is a hallmark of *M.tb* infection in susceptible mice and active TB patients, animal models suggest early recruitment of neutrophils into the lung, within the first two weeks, may be protective against *M.tb* infection[39,40]. In our study a mild BAL neutrophilia from D2 and increasing to D7 corresponded with a fall in APCs. In the previously reported BCG bronchoscopic instillation study in *M.tb*-exposed patients this increase in neutrophil frequency was seen at D3, which also coincided with a fall in macrophages[41]. In mice, infected neutrophils result in robust DC priming and migration to the lymph nodes, which may be what is occurring here[42]. In addition, our study demonstrated a fall in neutrophils in the blood from D7. A higher initial blood neutrophil count is protective against risk of developing

latent TB infection (LTBI) in healthy contacts of TB patients, while circulating neutrophils fall from 2-6 weeks compared with controls[32,43]. Furthermore, neutrophilia in the blood is associated with poorly controlled active TB disease, which resolves following TB treatment[44]. This might suggest that a robust immune response to a mycobacterial aerosol infection is associated with a fall in circulating neutrophils.

The evidence for a role for eosinophils in TB is limited. There is no evidence that eosinophils are infected with *M.tb* and the biological mechanism for eosinophil effector function remains unclear[14,45]. Depletion of peripheral eosinophils and a decreased systemic eosinophil to neutrophil ratio is associated with poor TB control in patients[46]. The increased circulating eosinophils and low circulating neutrophils seen in our study would be consistent with this finding, if these mechanisms contributed to BCG control in our volunteers. We saw eosinophils increase in the BAL at D7 compared with D2, however the high eosinophil count in D7 saline controls make these findings difficult to interpret. An early influx of eosinophils into the BAL following mycobacterial aerosol infection occurs in NHP and guinea pig models[47]. Eosinophils have been identified within granuloma, but the significance of these cells in the airway early after infection is unclear[46].

The influx of NK cells in the BAL in our study occurs much earlier than in mouse models, where NK cells appear 10-15 days post-mycobacterial aerosol infection[48]. In the human bronchoscopic instillation studies, BAL NK cell count did not change after PPD instillation at D2 or D3 in mycobacteria-exposed or naive patients[41,49,50]. However, in mycobacteria-exposed patients, there was a trend for an increased frequency of NK cells post-BCG instillation at D3[41]. In NHPs, NK cell frequency does not change following aerosol or IV BCG infection, but the earliest timepoint investigated was 2 weeks post-BCG[4].

Both NK cells and DURTs have the potential to produce Th1 cytokines early in infection, without the need for priming by APCs[37]. Furthermore, they have direct cytotoxic potential against mycobacteria-infected APCs[38,51]. Here we demonstrate that local mucosal NK cells and MAITs were expressing a Th1-Th17 gene signature and cytotoxic genes within 2-7 days following BCG infection. If these responses are shown to be protective, a vaccine which trains NK cells towards a protective cytotoxic or Th1 signature could be developed to confer protection against TB. Significantly, changes in NK cell and DURT frequency seen in the airways were not reflected in the blood, highlighting the importance of understanding local immune responses when designing vaccines.

The significance of the increase in proportion of CD4 + DURT subsets in the airways is unclear. Circulating CD4+ MAITs have reduced functional capacity compared with other subsets, even after in-vitro stimulation[52]. However, BAL MAITs displayed a robust effector function on gene expression, despite a high frequency of CD4+ MAITs.

The differences described in our study between animal models and human studies of *M.tb* infected subjects such as faster systemic antigen-specific IFN-γ+ responses may be because animal models and in-vitro human assays are imperfect models for in-vivo human responses[25-27]. In addition, many immune mechanisms described in human natural *M.tb* aerosol infections are based on observations of patterns in the systemic circulation, which can differ from local pulmonary responses. An alternate hypothesis for these different kinetics could be because BCG lacks virulence factors such as ESAT-6, postulated to be responsible for many immune evasion strategies[14,34]. The amount and frequency of mycobacteria that a TB contact is exposed to may differ from the aerosol dose given to our volunteers, which could impact the subsequent immune kinetics. Alternatively, the volunteers in this study may not have been mycobacteria-naïve, perhaps due to prior nontuberculous mycobacteria (NTM) exposure, thereby inducing faster antigen-specific responses. However, as volunteers were BCG-naïve, IGRA negative and were excluded if they had had prolonged residence in NTM-endemic areas, significant prior mycobacterial exposure is unlikely. This difference in kinetics is important

to understand and needs further work. Given a fast antigen-specific response appears critical for early mycobacterial clearance or control in animal models, understanding the underlying mechanisms which resulted in such rapid responses shown here could be key to developing a protective vaccine[35].

There is growing evidence highlighting the role of antibodies in protection against TB. We have previously demonstrated an association between decreased mycobacteria-specific IgG levels and an increased risk of developing TB disease[53]. In our model of BCG control, circulating IgG increased 2 months post-infection, though further work is required to confirm this result and any association with protection. The absence of detectable differences in antibodies in BAL fluid could be due to lack of sensitivity and the inter-subject variability in BAL fluid volume and concentration.

A limitation of this study is that the immune mechanisms described here may not be directly applicable to an *M.tb* infection due to the lack of certain virulence factors in BCG such as those encoded at the RD1 locus. While animal models and human studies have identified some immune evasion strategies employed by *M.tb*, it is unclear to what extent BCG utilises similar mechanisms[14,54]. Further, as BCG growth is controlled early and does not usually result in granuloma formation in immunocompetent individuals, this model cannot be used to describe the immune processes which contain *M.tb* within granulomas or those which trigger transformation from LTBI to active disease and hence cannot inform vaccine design aimed at preventing progression to disease. It would not be ethical to develop a controlled human infection model which resulted in disease in healthy people. In addition, while this is a model of BCG control, not all immune responses described may be contributing to this control. Immune responses seen here may be contributing to protective immunity, but many may be inconsequential or detrimental to mycobacterial clearance.

BAL cells were prioritised for immune analysis and hence only BAL fluid was used to culture and detect BCG. The BCG recovery rates from bronchoscopy described in this study are likely an underestimate. Subsequent work has shown that BCG recovery is higher when all BAL cells are used in the BACTEC MGIT (Harris et al., unpublished data).

We were limited in our flow cytometry analysis due to the numbers of BAL cells available. Some analyses were only carried out on a subset of volunteers who had more BAL cells retrieved. BAL cell count may be related to immune response and hence, missing values may have confounded some of our results. It also restricted our ability to draw robust conclusions around certain cell types such as the Donor Unrestricted T cells and polyfunctional T cells. Future studies with a larger sample size or more focussed analysis could study these cells in more detail.

Our scRNA-seq analysis was also limited by the small number of samples, with only three samples from the BCG group and three from the saline group sequenced at each time point. Future studies incorporating larger sample sizes to improve the statistical power and robustness of the scRNA-seq analysis are required to confirm the results.

This study was designed as an initial exploratory study, not powered for statistical significance and as such numbers, particularly in the saline control groups were low. For some markers, (such as SIGLEC8 to detect eosinophils, CD56 for NK cells and IFN-γ) the D7 saline control samples on flow cytometry stained with higher frequency than D2 control samples (this was a trend only due to low sample size). All three D7 saline control samples were processed, by chance randomisation, on the same day. Hence the difference in staining of some markers may have been due to a technical error. The results from D7 saline controls therefore need to be interpreted with caution. Alternatively, the increase in eosinophils, neutrophils and NK cells in day 7 saline volunteers (and BCG-infected volunteers) could be due to a reaction to the inhalation itself. Consistent with findings from

our previous study[18], volunteers who inhaled saline reported AEs such as cough and sore throat at a similar rate to BCG-infected volunteers.

In the absence of a virulent *M.tb* human challenge model, this aerosol BCG model is a useful tool to understand underlying immune mechanisms and to generate hypotheses for correlates of protection. Validation of immune mechanisms and correlates of protection identified using this BCG aerosol model, in parallel with *M.tb* animal studies and TB field trials, could help identify which mechanisms might also be protective against *M.tb*.

This model has been developed in a relatively homogenous population for logistical and safety reasons. Expansion of the model into a more diverse population, especially for those in *M.tb* endemic settings, is an important next step to improve the applicability of immune correlates.

We have previously described the development of a well-tolerated aerosol BCG controlled human infection model. Here, we used this model to demonstrate that aerosol BCG infection in *M.tb* and BCG-naïve UK adults elicits robust early innate and adaptive immune responses both at the respiratory mucosal surface and in the systemic circulation. Airway immune responses differed from that in the blood, highlighting the importance of human infection studies which deliver the infectious agent via a biologically relevant route and describe the subsequent responses at the infection site. This defined time point mycobacterial infection study has generated several immune pathways and markers for investigation. Once validated and elucidated for immune protective mechanisms, these could inform rational vaccine design and development.

## Methods

### Trial design

This single blind randomised controlled infection study complied with all relevant ethical regulations. The protocol and study documents were approved by the South Central Oxford A REC on 20 September 2018 (18/SC/0307). The study was registered with clinicaltrials.gov prior to commencement (NCT03912207, April 11, 2019) and was conducted at the Oxford University Hospitals NHS Trust and the Centre for Clinical Vaccinology and Tropical Medicine, University of Oxford, according to the principles of the Declaration of Helsinki and Good Clinical Practice. The safety of the study was overseen by an independent Safety Monitoring Committee.

The study was conducted in healthy, *M.tb* and BCG-naïve, UK adults using $1 \times 10^7$ CFU aerosol inhaled BCG Danish (AJ Vaccines) compared with inhaled saline controls, with bronchoscopy 2- or 7-days post inhalation (D2, D7). Volunteers with bronchoscopies at later timepoints (D14, D28, D56) were also subsequently enroled into this study. This clinical study was designed and planned from the beginning to analyse the innate and adaptive immune response induced in humans after a defined time point mycobacterial infection. BAL samples from the first two groups (with bronchoscopies at D2 and D7 post-infection, respectively) were analysed primarily for innate immunity. Samples from the subsequent 3 groups (bronchoscopies at D14, D28 and D56, respectively) were analysed primarily for adaptive immunity. It was not possible to do both innate and adaptive analyses for all volunteers due to the limited cell recovery from the BAL fluid. Flow cytometry panels as well as exploratory analyses were therefore different between the early phase of the study (Groups 1 and 2, with bronchoscopies at D2 and D7 respectively) focussing on innate markers, compared to the later Groups 3-5 (with bronchoscopies at D14, 28 and 56 respectively) which focussed on adaptive markers. Data from Groups 3-5 (bronchoscopies at D14, 28 and 56) cannot be compared directly with Groups 1-2 and will be presented in a subsequent manuscript.

The primary outcome of this part of the study (D2 and D7) was to define the early systemic and mucosal innate and adaptive immune responses induced following aerosol BCG infection. Secondary

outcome measures were to identify laboratory markers of the immune response that correlated with protection as defined by a PBMC Mycobacterial Growth Inhibition Assay (MGIA). Tertiary outcomes were to describe the human clinical response to aerosol BCG challenge.

Volunteers aged 18-50 years residing in or around Oxford were recruited using advertising. Volunteers were screened following formal written consent and excluded if they were smokers, had previous or current respiratory disease, were pregnant or breastfeeding, or had a clinically significant medical condition. Mycobacterial pre-sensitisation was minimised by excluding volunteers who were Interferon-Gamma Release Assay (IGRA) positive, had clinical evidence of prior BCG vaccination or had resided in a tropical country for more than 12 months.

### Infection agent

Each vial of BCG Danish contains a dried lyophilised powder of $2\text{-}8 \times 10^6$ CFU live attenuated Danish strain 1331 *Mycobacterium bovis* BCG, manufactured under Good Manufacturing Practice conditions by AJ Vaccines, Denmark. A median of $5 \times 10^6$ CFU/vial was assumed for dosing calculations. The challenge agent was reconstituted with the supplied solvent, concentrated by combining vials and made up to 1 ml with 0.9% sterile sodium chloride, to achieve $1 \times 10^7$ CFU/ml. Controls inhaled 1 ml sterile 0.9% sodium chloride (Demo SA pharmaceuticals). BCG or saline was aerosolised using the Omron MicroAir U22 ultrasonic mesh nebuliser (Omron Healthcare UK, Ltd., Milton Keynes, UK).

### Trial conduct

Volunteers were sequentially enroled by the study investigator into either Group 1 ($n = 13$) or Group 2 ($n = 13$) and randomised 10:3 to inhale either BCG or saline by variable randomisation using sequentially numbered sealed envelopes, prepared by an independent statistician. Volunteers were blinded to the inhaled agent. Volunteers underwent fibreoptic bronchoscopy at either 2 days (D2, Group 1) or 7 days (D7, Group 2) post-infection.

The bronchoscopy was performed by a consultant respiratory physician under standard Oxford University Hospitals NHS Trust procedures, using optional intravenous sedation (Midazolam and Fentanyl) according to volunteer preference, and local anaesthetic spray (2% lidocaine) above and below the vocal cords. The macroscopic appearance of the airways was recorded before bronchoalveolar lavage (BAL) was obtained from the right middle lobe (medial segment) using 150 ml of 0.9% sodium chloride. Up to six endobronchial biopsies of the subcarinal region were then taken, as tolerated. The bronchoscopist was blinded to inhaled agent.

### Safety

Safety was assessed by collecting frequency and severity of solicited and unsolicited adverse events (AEs) throughout the 6-month study period. Expected respiratory AEs (cough, sore throat, tickly throat, wheeze, dyspnoea, sputum production, haemoptysis, chest pain, chest tightness) and systemic AEs (fever, feverishness, fatigue, malaise, headache, myalgia, arthralgia, nausea) were solicited from subjects using an electronic diary card for 14 days post-infection (twice daily for the first 2 days then once daily in the evening), and reviewed at each clinic visit. Volunteers were supplied with a digital thermometer. Induced sputum was collected for BCG detection at 3- and 6-months post-infection, following 20 min of 3% hypertonic saline inhalation, from volunteers who received aerosol BCG. Stopping criteria were any related serious adverse event or two or more participants having the same related grade 3 adverse event beginning within 2 days after challenge and persisting for more than 48 h.

Blood biochemical and haematological parameters were measured at baseline and D28 post-infection. Vital signs were measured at all clinic visits, and spirometry was measured at visits to 2 months post-infection only, unless clinically indicated. The transfer capacity of uptake of carbon monoxide (TLCO) was measured at baseline and at D7 post-infection. Urine β-HCG analysis was performed on day of infection and at bronchoscopy in volunteers of child-bearing potential.

### Immunology

**BCG detection.** The infection dose was verified by plating serial dilutions from the reconstituted vaccine vials onto solid Middlebrook 7H11 agar (Sigma-Aldrich) and the more sensitive BACTEC™ MGIT™ system was used to culture BCG from BAL and sputum, as previously described[18]. Briefly, a BACTEC™ MGIT tube (Becton Dickinson, UK) was prepared for each BAL or sputum sample by supplementing the Middlebrook 7H9 with 800 μL of BBL MGIT OADC (oleic acid, albumin, dextrose, and catalase) and PANTA (polymyxin B, amphotericin B [AMB], nalidixic acid, trimethoprim [TMP], and azlocillin) mixture (Becton Dickinson, UK), as recommended by the manufacturer. After removal of the cells for flow cytometry, BAL fluid (BALF) was centrifugated at $3000 \times g$ for 17 min, supernatant was decanted into a new tube and frozen at $-80\,°C$. The remaining pellet was then decontaminated using the BBL® MycoPrep™ Specimen Digestion/Decontamination Kit (Becton Dickinson, UK) according to the manufacturer's instructions. The decontaminated samples were washed in PBS, centrifuged at $3000 \times g$ for 17 min, the supernatant discarded, and the pellet resuspended in media from the corresponding supplemented MGIT tube and returned to the tube. Tubes were placed on the BACTEC™ MGIT instrument (Becton Dickinson, UK) and incubated at 37 °C until the detection of time-to-positivity (TTP) by fluorescence. TTP is inversely associated with the number of CFU present in the sample.

**HAIN™ sequencing.** The BACTEC™ MGIT™ system was used to detect live mycobacteria. The growth was then confirmed as BCG by HAIN™ genotyping. DNA, extracted from the MGIT™ tubes was amplified and genotyped using the HAIN lifescience GenoType MTBC VER 1.X as per the manufacturer's instructions as previously described[18]. The Hain MTBC kit detects *M. tuberculosis*, *M. bovis* (including BCG), *M. canettii*, *M. africanum*, and *M. microti*. DNA was extracted from all MGIT™ tubes regardless of whether they were positive or negative, to ensure detection of dead BCG in the MGIT™ negative tubes.

**Bronchoalveolar lavage and sputum processing.** BAL and sputum were collected on ice and processed within 2 h as previously described[18]. BAL cells were separated from the BALF and used for exploratory immunological analyses and only the supernatant (BALF) was used for BCG detection.

**Stimulation of BAL and Whole Blood (WB) for intracellular cytokine staining.** Fresh WB was stimulated at baseline (D0), D14, D28 and D56, and at either D2 (Group1) or D7 (Group 2)[55]. BAL cells were resuspended to achieve a concentration of $1 \times 10^6$ live lymphocytes/ml. 1 ml BAL or WB were stimulated overnight with BCG Pasteur ($1.2 \times 10^6$ CFU/ml; Aeras), and otherwise as previously described[18].

**Immunohistochemistry staining of endobronchial biopsies.** Biopsies were collected and immersed in 10% neutral buffered formalin, then embedded into paraffin wax prior to cutting. Sequential 4-μm sections were stained with Hematoxylin and Eosin (H&E), Ziehl-Neelsen (ZN) and immunohistochemistry (IHC) to label cell markers (Supplementary Table 2). Stained slides were evaluated by a qualified veterinary pathologist blinded to infection and timepoint and were randomised prior to examination to prevent bias. H&E, ZN and IHC-stained slides were digitally scanned in a Hamamatsu S360 digital scanner and examined using ndp.view2 software (v2.8.24). Digital slides were evaluated with image analysis software Nikon NIS-Ar to

calculate the area of the tissue section positively stained against the different cell markers, using appropriate thresholds for each marker and Regions of Interests (ROIs).

**Mycobacterial growth inhibition assay (MGIA).** A total of 500 CFU BCG Pasteur and $3 \times 10^6$ PBMC in a total volume of 480 µl MGIT media with 120 µl autologous serum (matched to time-point) per well were incubated for 96 h at 37 °C, then lysed with 500 µl sterile water. Supernatant was discarded and the lysate was transferred to MGIT tubes supplemented with 800 µl of BBL MGIT OADC/PANTA, as recommended by the manufacturer. Tubes were incubated at 37 °C in the BACTEC 960 machine until Time-To-Positivity (TTP) was detected by fluorescence. Duplicate supplemented BACTEC MGIT tubes were inoculated with the same volume of mycobacteria as the samples and incubated on the BACTEC 960 machine from day 0, to create direct-to-MGIT viability control tubes.

**Ex-vivo enzyme-linked immunospot (ELISpot) assay.** Fresh PBMC were used in an ex-vivo IFN-γ ELISpot assay at baseline, D2 (Group 1 only), D7, D14, D28, D56, D84 and D168[56]. Briefly, cells were stimulated in triplicate wells at $3 \times 10^5$/well with 20 ug/ml of PPD (AJ Vaccines, Denmark), 10 ug/ml Staphylococcal Enterotoxin B (SEB, positive control; Sigma) or left unstimulated as a negative control for the assay. Background (unstimulated) subtracted PPD-specific responses are presented as Spot Forming Cells (SFC) per $1 \times 10^6$ PBMC.

**Enzyme-linked immunosorbent assay (ELISA).** ELISAs, measuring PPD-specific IgG and IgA, were performed on serum taken at baseline and D2, D7, D14, D28, D56, D84 and D168 post-BCG infection or saline inhalation, as previously described[18]. Briefly, ELISA plates (NUNC 442404) were coated with 50 µL/well of 5 µg/ml PPD and incubated overnight at 4 °C. Serum samples were diluted 1 in 50. Standards were added in duplicate, and plates incubated at room temperature for 2 h. Anti-human goat IgG or IgA (y-chain specific)-alkaline phosphatase antibody (Sigma-Aldrich) was added (1 in 1000 dilution). Plates were developed by adding 100 µL/well of p-Nitrophenyl Phosphate, Disodium Salt (pNPP) substrate (Sigma-Aldrich) and were read at 405 nm using Gen5 software (v2.0.7, BioTek).

Prior to running ELISAs on BALF, phospholipid concentration was measured by means of a phospholipid assay kit (Sigma-Aldrich) and the BALF samples concentrated to 0.5 mg/ml phospholipid, as previously described[18], except that Amicon Ultra-15 Centrifugal Filter Units (10KDa) were used due to a larger volume of BALF stored. BALF antibody arbitrary units (AU) were then multiplied by 2 in order to present BAL antibody per 1 mg/ml phospholipid.

**Flow cytometry.** Four separate flow cytometry panels were used to detect cell types and antigen specific cytokine production (Supplementary Note 2). BAL cells were allocated sequentially starting at Panel 1. The number of panels used for each volunteer was limited by the amount of BAL cells available. Blood samples were stained with all four panels at baseline, D14, D28 and D56, and D2 (Group 1) or D7 (Group 2).

For phenotype staining, cells were stained for viability with Live/Dead stain (except for lysed fixed WB cells) followed by surface staining. For panels analysing antigen presenting cells, cells were incubated with Fc-receptor blocker (Biolegend) alone prior to staining with antibody mix combined with Fc blocker. MAIT cells were detected by incubating with tetramers (MR1 5-OP-RU (MHC class I-related protein 1 5-(2-oxopropylideneamino)-6-D-ribitylaminouracil); MR1 control 6-FP (6-formylpterin), NIH Tetramer Core Facility) prior to surface staining for other markers[57]. For detection of intracellular cytokine responses, after surface staining, cells were permeabilised and stained intracellularly.

All stained cells were fixed with 1% paraformaldehyde, acquired within 24 h on an LSR Fortessa v.2 Std X20 flow cytometer using BD FACSDiva 8.0.2 and analysed on FlowJo (BD) v10.

**Single cell RNA sequencing (scRNA-seq).** BAL cells from three BCG-infected volunteers from each group and all control volunteers were processed for scRNA-seq. Cells in Group 1 were first incubated with CD3+ microbeads (Miltenyi Biotec), to enrich for CD3+ cells to ensure there were sufficient cells to describe T cell subsets. However, as the BAL was sufficiently enriched for T cells to detect T cell subsets without the need for CD3+ enrichment, Group 2 cells were not incubated with CD3+ microbeads to minimise cell handling. Incubation results in 30% epitope (light-touch) attachment per cell, allowing ongoing functionality and availability for TCR sequencing and therefore sequencing results between the two groups should not have been affected[58]. Furthermore, a head-to-head comparison was made to compare sequencing by both methods, which found no difference in cell proportions. Despite this, cell proportions identified by scRNA-seq have not been reported here, with results from scRNA-seq limited to reporting detection of cell types and differential gene expression analysis.

Single cell RNA-seq libraries were created from BAL cells as per manufacturer's instructions using 10x Genomics single cell RNA 5 prime kits (v1.1), reference genome GRCh38 (human38 (Genome Reference Consortium)). The target number of cells per channel was 5000 and two channels were used per sample (with the pooled count of 10,000 cells sequenced). Libraries were sequenced on a Novaseq 6000 S4 flowcell (150 paired-end configuration). Accepted minimum sequencing depth was 30,000 mean reads per cell and 50% sequencing saturation.

**scRNA-seq data alignment and sample aggregating.** All analysis was performed on R version 4.1.2 and Python version 3.8.19.

Barcode assignment, Unique Molecular Identifier (UMI) tagging, and sequence alignment for the BAL samples were conducted using the Cell Ranger count pipeline (v6.0.1), employing the human reference genome GRCh38 (GENCODE version 32/Ensembl 98) provided by 10x Genomics[59].

Potential contamination by ambient RNA was removed using the SoupX package (v1.6.1)[60]. Data processing and doublet exclusion were then executed using the Scanpy toolkit (v1.9.8)[61]. Cells were excluded if they had fewer than 200 genes detected, more than 10% mitochondrial gene expression, or if their doublet score surpassed the 0.75 quantile by 1.5 times of the interquartile range (IQR), as determined by the Scrublet (v0.2.3)[62]. After filtering, a total of 91,958 cells were left for downstream analysis.

**Dimensionality reduction and clustering.** For dimensionality reduction, the gene-cell matrix underwent normalisation and log transformation. The Cell Ranger algorithm was used to identify the top 2000 variable genes. Data was standardised to unit variance before PCA was executed on these genes. The inflection point of the explained variance was found[63] and was used to determine the number of principal components (PCs) to retain for the downstream analysis.

Harmony (harmonypy, v0.0.6) was applied to the PCA outputs for batch effect correction[64]. Cosine distances between cells were used to construct a neighbourhood graph. Cell clustering within this graph was performed with the leiden algorithm (leidenalg, v0.8.10)[65]. Uniform Manifold Approximation and Projection (UMAP, umap-learn v0.5.3) was used for visualisation of the clustered data[66].

**Marker identification and cell annotation.** Student's t-test in the Scanpy was used to discern marker genes for each cluster against all others. Cell clusters were manually annotated using the literature-supported marker genes. Macrophages, T cells and NK cells in the BAL

samples were reintegrated separately following similar processing as described above.

**Differential gene expression analysis.** The expression values of each gene across all cells within a given cell type and sample was aggregated to construct a pseudo-bulk expression matrix. Only genes that were expressed in more than 5% of all cells were included in the expression matrix. DESeq2 package (v1.34.0) was used to identify DEGs between volunteers challenged with BCG at different time points and saline groups within each cell type[67]. Genes with an adjusted *p*-value of less than 0.05 were considered DEGs. Over-represented BTMs and Hallmark gene sets in the upregulated and downregulated DEGs were identified using the clusterProfiler package (v4.2.2)[68]. Gene sets with an adjusted *p*-value of less than 0.05 were considered enriched. Gene sets that were over-represented both in the upregulated and downregulated DEGs were excluded. GO enrichment analysis[69] was performed using the enrichGO function in clusterProfiler package (v4.2.2)[68]. DEGs were grouped into upregulated genes and downregulated genes, and then enriched separately. Enriched terms of upregulated genes per cell type were reported.

**T cell signature and PPD-response signature score.** Gene signature scores were calculated for each cell type and sample using the pseudo-bulk expression matrix. The pseudo-bulk expression matrix was normalised and $log_2$-transformed using the estimateSizeFactors and counts functions in DESeq2 (v1.34.0)[67]. The gene signature score for each cell type and sample was calculated as the average expression level of the genes within the gene signature.

**Statistical analyses**

The sample size of 10 per BCG group was determined based on our previous experience with phase 1 exploratory clinical studies and chosen with the aim of detecting substantial differences in the primary outcome measure between the two groups, while also being a feasible number to enrol. The sample size was not determined with the aim of achieving statistical significance. Statistical analysis was performed using GraphPad Prism Software v.9 and 10, SPSS v.25 and R v.4. Comparisons between timepoints were made against baseline only unless otherwise stated. A Shapiro-Wilk test was performed for normality testing. Comparisons were made between groups using two-tailed Wilcoxon signed rank test for paired data; or two-tailed Mann-Whitney for unpaired data. Correction for multiple comparisons was done using the Friedman with Dunnett's test or Dunn's multiple comparisons test. An AUC analysis of deviation from baseline was used to compare overall spirometry values between groups after infection. Mixed methods Anova with Bonferroni correction was used for analysing CD4/CD8 subsets in Donor Unrestricted T cells. Single cell sequencing analysis was performed as outlined above. Missing data was as outlined. Collection of blood at certain timepoints was not possible due to COVID-19 pandemic restrictions which was unrelated to the study design or volunteer characteristics, and was assumed random for analysis. Other missing data points, due to insufficient BAL cells to undertake all the exploratory analysis on all individuals, may not be random.

**Reporting summary**

Further information on research design is available in the Nature Portfolio Reporting Summary linked to this article.

## Data availability

The raw data and processed data of the scRNA-seq generated in this study have been deposited in the Gene Expression Omnibus database under accession code GSE282132. The gene signature for T1-T17 Subpop 1 and 2 is available at https://ars.els-cdn.com/content/image/1-s2.0-S1074761322001753-mmc5.xlsx. The gene signature for PPD response is available at Table S8 of https://www.science.org/doi/10.1126/sciadv.adq8229#supplementary-materials. Source data are provided with this paper in the Source Data File. For all other data, any reasonable request for raw or analysed data will be reviewed by the study team, and a response can be expected within 14 days. The data generated in this study are subject to patient confidentiality, and the transfer of data or materials will require approval from the sponsor. Any shared data will be de-identified. Requests should be made to HMcS. Source data are provided with this paper.

## Code availability

No custom software was generated for this manuscript. The software used for all analyses is listed in the relevant sections of the Methods.

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

## Acknowledgements

This research was funded in whole, or in part by the Wellcome Trust. H.McS. is a Wellcome Trust Investigator (grant code WT 206331/Z/17/Z). For the purpose of open access, the author has applied a CC BY public copyright licence to any Author Accepted Manuscript version arising from this submission. This research was funded/supported by the National Institute for Health Research (NIHR) Oxford Biomedical Research Centre (BRC). The views expressed are those of the author(s) and not necessarily those of the NHS, the NIHR or the Department of Health. J.L.M. received funding from the British Federation of Women Graduates (FfWG) and The University of Oxford (Goodger & Schorstein Research Scholarship). M.S. was supported by the VALIDATE Network, which is funded by the UK MRC/BBSRC (MR/R005850/1) and the Bill and Melinda Gates Foundation (INV-031830). T.S.C.H. was supported by a Wellcome Career Development Fellowship (211050/Z/18/Z). The following reagents were obtained through the NIH Tetramer Core Facility: MR1 5-OP-RU; MR1 control 6-FP. The MR1 tetramer technology was developed jointly by Dr. James McCluskey, Dr. Jamie Rossjohn, and Dr. David Fairlie, and the material was produced by the NIH Tetramer Core Facility as permitted to be distributed by the University of Melbourne. We would like to acknowledge the Oxford Vaccine Centre (OVC) Biobank. We thank Nicola Williams from the Nuffield Department of Primary Care, The University of Oxford, for conducting the study randomisation and providing statistical advice. We are grateful for the advice and oversight provided by the Safety Monitoring Committee, Professor Hassan Mahomed, Professor Stephen Gordon, and Professor Francesca Little, and the invaluable contribution by CCVTM clinical staff, including Michelle Fuskova, Ian Poulton, Megan Baker and Colin Larkworthy.

## Author contributions

H.McS., J.L.M., and I.S conceived the study. J.L.M., I.S., M.S., S.A.H., H.M., R.W., M.P.P.A., S.L., E.H., F.J.S., R.T., and H.McS. developed the methods. J.L.M., I.S., M.S., S.A.H., R.W., M.P.P.A., S.L., F.J.S., and H.McS. conducted the formal analysis and accessed and verified the data. J.L.M., R.L.R., F.R.L., C.M., I.C.P., R.P.D., T.S.C.H., H.B., and H.McS. did the investigation. J.L.M., I.S., S.A.H., S.L., and H.McS. wrote the original draft of the manuscript, which was reviewed and edited by all authors. H.McS. and I.S. supervised the project and H.McS. secured the funding. All authors had full access to all the data in the study and have final responsibility for the decision to submit for publication.

## Competing interests

The authors declare no competing interests.

## Additional information

Julia L. Marshall ®[1,2,5], Iman Satti[1,5], Mirvat Surakhy ®[1], Stephanie A. Harris[1], Hazel Morrison[1], Rachel E. Wittenberg ®[1], Marco Polo Peralta Alvarez ®[1], Shuailin Li ®[1], Raquel Lopez Ramon[1], Emily Hoogkamer[1], Francsico Javier Salguero[3],

**Fernando Ramos Lopez** ⓘ **[1], Celia Mitton[1], Ingrid Cabrera Puig[1], Rebecca Powell Doherty[1], Rachel Tanner[1],
Timothy S. C. Hinks** ⓘ **[4], Henry Bettinson[4] & Helen McShane** ⓘ **[1]** ✉

[1]The Jenner Institute, Nuffield Department of Clinical Medicine, University of Oxford, Oxford OX3 7DQ, United Kingdom. [2]The Peter Doherty Institute for
Infection and Immunity, The University of Melbourne, Melbourne, Australia. [3]UK Health Security Agency, Porton Down, Salisbury SP4 0JG, United Kingdom.
[4]Oxford Centre for Respiratory Medicine, Nuffield Department of Clinical Medicine, The University of Oxford, Oxford OX3 7DQ, United Kingdom. [5]These
authors contributed equally: Julia L. Marshall, Iman Satti. ✉e-mail: helen.mcshane@ndm.ox.ac.uk

