## [Transparent Peer Review file · Nature Communications]

Early mucosal responses following a randomised controlled human inhaled infection with attenuated *Mycobacterium Bovis BCG*

Corresponding Author: Professor Helen McShane

Version 0:

Reviewer comments:

Reviewer #1

(Remarks to the Author)

In the current study the authors utilized a BCG human infection model to define initial innate and adaptive immune responses. I commend the authors on an excellent body of work. I have a few comments for the authors to consider:

1. It is unfortunate that the sample size was limited, which impacted the ability to draw robust conclusions about MAIT, iNK, and polyfunctional T-cells. The authors do acknowledge this as a limitation. Future studies with larger sample sizes or optimized methods to obtain more BAL samples could potentially address these gaps. The authors would need to expand on this in the limitations section.
2. In figure 2 it is not clear how the authors phenotyped the APCs? I assume it is a combination of macrophages and DCs. Did the authors look at pDCs and cDCs? Can they draw any conclusions from this data. It does appear as if the cDC population increases in the BCG challenged versus placebo participants after 2 days post-challenge for the single cell sequencing data. The increase in the cDC population in response to BCG challenge is noteworthy and could suggest a differential role or response of cDCs compared to other APCs. The authors should clarify their methods for phenotyping these cells and discuss the implications of their findings in relation to the cDC increase.
3. It seems strange that the authors did not look at T-cell memory responses in the current study. The authors need to address this gap or justify why it was not included in their study.
4. What was the activation status of the neutrophils in the BAL versus the blood pre- and post-challenge. Neutrophils in the BAL could show differential activation or functional responses compared to those in the peripheral blood, which could be relevant for understanding the local immune response in the lungs.
5. No differences were observed for the MGIA assay with PBMCs post BCG inhalation versus baseline. Could this be explained by the differences in the memory immune responses in the lung versus the peripheral blood? Did the authors attempt to use BAL cells in these assays? Maybe a more robust assay would have been to use the mycobacterial stasis assay used by other groups by counting CFUs? Can the author's comment.
6. HAIN sequencing was used to detect mycobacteria in the BAL. As the authors are aware the assay would not be able to discriminate between live and dead bacteria in the lung. A positive signal may simply have been generated by free genomic material in the lung. The authors need to comment on this in the manuscript.
7. In the results section, line 199, the incorrect figure is referenced. "Fig 4D" should be "Fig 4E".

Reviewer #2

(Remarks to the Author)

In their manuscript entitled "Early immune responses in the lung and blood following a randomised controlled human inhaled infection with aerosolised attenuated *Mycobacterium Bovis BCG* in healthy, BCG-naïve, UK adults" the authors further characterize the BCG controlled human infection model, describing in great detail both the systemic immune response and the respiratory response found in BAL fluid and biopsies. While the ability to follow the participants according to the original protocol was constrained by the COVID-19 pandemic, there is significant new information in this manuscript. As had been previously reported, the inhaled BCG at the dose given was well tolerated and immunogenic. The participants who received BCG had more significant symptoms than the controls who inhaled saline. The authors found that the immune response to inhaled BCG was faster than what had previously been reported for intradermal administration, or what was

seen in natural mycobacterial infection. While there was not much of an immune signal after 2 days in either the blood or BAL, there was a significant immune response to the BCG by day 7, including an IFN gamma response. The paper is clear and easy to read.

My comments/questions are below

1. In the previous study (Satti, Lancet 2024), the dose of the challenge was lower than anticipated due to both the variability of BCG in the vials- and also the loss of some of the dose in the nebulizer. In this manuscript, the authors state the BCG when reconstituted was 0.5 Log lower than stated on the label. How was this controlled for in the current study? Did the dose given take that lower level into account?

2. Results, Enrolment, Line 81-82:

The authors state: "Fourteen follow up visits (D56, D84, D168; all Group 2 volunteers) were conducted as telephone visits". This sentence is confusing. Were all of the Group 1 D56-168 visits remote? In addition to the Group 2 visits? But the Group 2 volunteers had their Day 7 BALs done?

3. Only 60% of volunteers who were given inhaled BCG had viable organisms recovered on BAL on day 2 after challenge (and an even lower percentage on day 7). Could this be due to inadequate challenge dose not leading to an infectious "take" in those who had negative cultures?

4. Results section 4. Line 116.

It is unclear to this reviewer why the PBMC Mycobacterial Growth Inhibition assay did not demonstrate the ability of cells to control the BCG.

5. Figure 2: Eosinophils, and possibly neutrophils and NK cells

In looking at the figure, it seems that the eosinophils (and possibly neutrophils and NK cells?) are higher on Day 7 vs. Day 2 regardless of whether they were challenged with BCG or inhaled Saline. In fact, it is significantly higher in the saline control group than those challenged, albeit the numbers are small. Could it not be just a reaction to the inhalation rather than to the BCG challenge?

The only panel which seems to be convincingly different between the control and challenge on day 7 is the APCs.

6. Figure 3: Panels G, H, I

What are the colors? There seems to be a missing legend.

Minor comment:

Methods 2.2 Bronchoalveolar lavage and sputum processing

Line 524: What is the abbreviation BALF?

Reviewer #3

(Remarks to the Author)

A descriptive study of immune changes in bronchoalveolar lavage (BAL) and systemically following aerosolised Bacille Calmette Guerin (BCG) challenge in immunologically naïve humans is presented. Follow up was disrupted by the COVID-19 epidemic.

Results of interest include that

1. Inhalation of what appears quite a substantial dose of bacteria (intended as 1×10^7) was in general well-tolerated
2. It was difficult however to recover live bacteria from subsequent BAL being found in 6/10 on day 2 and 2/10 on day 7.
3. Distinct changes in the cellular composition of BAL were induced by BCG mostly clearly of neutrophils, NK and DURT cells and both CD4 and CD8 positive lymphocytes. Such changes were not as prominent in peripheral blood.
4. By day 7 in BAL the frequency of CD4 and CD8 (and DN and DP) positive cells that were positive for Interferon (IFN)-gamma upon restimulation with BCG increased
5. A marked transient increase in the frequency of peripheral cells producing IFN-gamma on restimulation was observed at day 7
6. IgA and IgG antibody responses to BCG in both BAL and serum were not pronounced.
7. Single cell RNA sequencing of BAL from 6 donors (3 controls, 3 BCG) at varying timepoints confirmed BCG activation of both innate and adaptive immune pathways

The work appears well-performed and is technically challenging. Some of the results are intuitive and in line with what might have been expected although the ostensible induction of an acquired T cell response both in BAL and peripherally in naïve donors is, as the authors say, quite interesting and might not have been so predictable given results of studies in animals challenged with virulent strains of mycobacteria.

The authors may consider the following

1. The study was conducted over 4 years ago. It is acknowledged the authors allude to disruption due to COVID-19 which is understandable but also mention intent to bronchoscope at later time points (D14, D28, D56). Surely these timepoints have

been collected yet the intent to present data is described as being for a future paper

2. Flow cytometric results throughout are reported as percentage so it is inevitable some subsets will appear to fall if others rise. A supplementary table stating the actual numbers of cells recovered at BAL in each donor (or a median) would help put these percentages in perspective even if it is accepted absolute values would be difficult to calculate because of variable recovery and volume of BAL fluid.

3. The authors acknowledge that the inoculum did not turn out as high as expected, and also that the aerosol delivery will distribute this widely throughout the lung which in turn will undoubtedly affect recovery of bacilli at subsequent bronchoscopy. Nevertheless, widespread infection with a high dose of an avirulent Mycobacterium is counter to widespread belief that natural human infection by *M. tuberculosis* may arise from very few virulent bacilli that establish a localised lesion. The authors do discuss this but it is a limitation. Notwithstanding that the HAIN was positive, the rate of disappearance of viable bacilli is quite striking especially in ostensibly unsensitised donors (see below). Overall is what is set up in this model a transient very mild pneumonitis rather than a localised granulomatous focus?

4. The very substantial but transient peripheral response is also interesting. The authors discount that this may have arisen via prior sensitisation because of careful selection of donors. However the day 0 peripheral ELISpot response to BCG Danish is not entirely negligible especially bearing in mind the scale of figure 4E. An aside is why was BCG Danish chosen to restimulate when the challenge was BCG Pasteur?

5. In Figure 4D reported responses are heavily influenced by outliers

6. The reason for missing data in Figure 3 panel 1 presumably relates to the number of cells available and should be stated in the legend

7. Given expansion of neutrophils why do these appear so infrequent in the sc RNA seq?

8. With respect to legends these excessively emphasise statistical testing and as such are dense and do not really explain what was done. It may be the journal style but I suggest stating the statistical tests on the last line of each legend and concentrating on a brief description of the data.

9. In the methods line 527-528 it is stated BAL cells were resuspended to achieve 1×10^7 lymphocytes/mL. Do the authors mean lymphocytes or all cells?

10. Spell out OP-RU and 6-FP in methods

11. Line 796 how can the study have been funded in whole, or in part, by Wellcome Trust?

12. Reference 44 is now published as *J Exp Med.* 2021 Oct 4;218(10):e20210469. It is not, as stated, a review. The lead author of reference 36 was Brill, KJ and Kaufmann, SHE was not an author at all.

13. Other issues with referencing include lack of italics for species names (8,14,22,29,30,32-34,36,40,43,45,46-48,52); page numbers missing (2,3); year missing (18).

Reviewer #4

(Remarks to the Author)

Summary

This manuscript aims to describe human in vivo local and systemic immune responses to aerosolised BCG vaccination. It is the first such study to do so. It evaluates cellular responses by flow cytometry and single cell RNA sequencing, some evaluation of antibody responses, and uses an ex vivo peripheral blood mononuclear cell-based mycobacterial growth inhibition assay (MGIA) as surrogate measure of protective immunity. The opportunity for novel insights is driven predominantly by bronchoalveolar lavage sampling after aerosolised vaccination, but the data presented are limited to day 2 or day 7 BAL sampling time points aiming to focus on 'early' responses in the present report. For the most part, the interpretation of the data is critically limited by inadequate statistical power to make confident conclusions. The most robust finding is evidence for enrichment of T cells and antigen-specific Th1 responses by flow cytometry of BAL samples that is supported by evidence for increased TCR and IFN-signalling in single cell RNAseq data, along with increased frequency of circulating antigen specific T cell responses at day 7 identified by ELISpot. I don't find this a particularly significant advance, but the fact that it is being reported for the first time in this context has some incremental value, albeit preliminary in nature.

Major comments

There is no evidence of time-dependent increase in protective immunity generated by the aerosolised BCG vaccine in the MGIA data compared to pre-vaccination samples. Data from the control group receiving aerosolised saline are not presented- I think they should be included. However, it is not clear whether we should interpret this as evidence for lack of vaccine efficacy overall, lack of systemic protection in contrast to the potential of local protection in the respiratory tract, or simply that the MGIA is not a suitable outcome measure for protective immunity. I am aware that the authors have previously reported MGIA as evidence for protective immunity in vaccinees receiving conventional intradermal BCG, but is not included as a comparator group in the present study, so difficult to know what to make of the data.

The fact that no measure of vaccine mediated 'protection' is shown, means that no assessment of immune correlates of protection are possible. Hence the study cannot distinguish between immune responses that might be protective, inconsequential, or even harmful. The idea that any or all of the vaccine induced responses detected are likely to be contributing protective immunity is wholly speculative.

The statistical assessments of immune cell frequencies in BAL samples are muddled and misleading. First, the use of percentages to quantify cellular infiltration is uninterpretable- do these represent increases in absolute cell number or changes in proportions? The authors should really present some measure of changes in absolute number standardised by sample volume or another constant (eg epithelial cell frequency). Second, given the study design includes saline control

recipients, the principal analysis should really focus on whether there are changes attributable to BCG compared to saline controls at the same time point. For example, the authors state “There was an increase in frequency of eosinophils, neutrophils, NK cells and the cytotoxic CD16+ NK cell subset in the airways of volunteers infected with aerosol BCG 123 at D7 compared with D2 or saline controls (Fig. 2)”. In fact, there is no evident statistically significant increase between D7 BCG and saline controls for eosinophils, neutrophils or NK cells. Third, reporting of statistically significant findings should only be reported after multiple testing corrections. If the authors limit their testing to comparisons with contemporary saline controls, then they may incur less multiple testing penalty than performing all pairwise testing. This critique applies to all flow cytometry panels in Figs 2, 3 & 4, the ELISpot data in Fig 4, and the PPD-specific antibody measurements in Fig 5.

It's a shame the authors chose to undertake T cell bead enrichment of the day 2 samples, which as they acknowledge, precluded any comparison of differences in cluster abundance between these time points. The UMAP figures presented in Fig 6A do not provide any particular analytical value. The module and pathway enrichment analysis of differential gene expression between aerosol BCG vaccinees and saline controls is helpful, and supports the finding of increased IFN γ and TCR signalling at day 7 following BCG. However, it's not clear whether the DGE analysis is at the level of individual donor (ie pseudobulked cluster data for each individual) or individual cell level. I suspect it is the latter, in which case we can't assess to what extent the findings are generalisable between BCG vaccine recipients, or the results are driven by one or a subset of individuals.

On a separate note, biological inferences made on the basis of pathway enrichment derived from DGE alone remain rather speculative – those that the authors wish to highlight as biologically interesting really require orthogonal validation – for example by independent/experimentally derived gene expression signatures that represent the biological process or immune pathway of interest.

Given the lack of statistical power and other weaknesses in the analyses presented, I find much of the points highlighted in the discussion overly speculative.

Minor points

The study participants are substantially enriched for young white females. This further undermines the lack of power to make generalisable observations.

The authors should discuss the fact that BAL measurements are limited by sampling error. It's a shame that they did not include at least two separate lobes/segments of lung to lavage so they can evaluate some measure of intra-individual consistency of these measurements. This is true not only for the microbiological measurements as well as the immunological assays. For example, how confident are they that yield of viable bacteria is not systematically different at different site in the lung?

The reference to BCG-specific monocytes (line 194) is particularly odd/misleading concept as there is no mechanism by which monocytes can be 'specific' for BCG. I assume they mean BCG-reactive monocytes.

Version 1:

Reviewer comments:

Reviewer #1

(Remarks to the Author)

I commend the authors on an excellent study. The authors acknowledged their limitations when using BAL cells. I am satisfied with the authors responses and the amendments to the manuscript.

Reviewer #2

(Remarks to the Author)

The authors have satisfactorily addressed my questions in the submitted revised manuscript. I have no additional questions, except that one comment from another reviewer re: italicizing genus/species names in the references does not seem to have been done.

Reviewer #3

(Remarks to the Author)

The authors have made a conscientious effort to address my comments and have introduced changes in the manuscript

Reviewer #4

(Remarks to the Author)

I reviewed the revised manuscript focussing on the authors' response to my previous comments. In general, it seems that the authors acknowledge most of the critique, but have not been able to address any of the fundamental weaknesses in their study. The manuscript offers some preliminary descriptive analysis of early immune responses in BAL samples following

aerosolised BCG vaccination, but fails to offer any new biological insights.

I welcome the addition of saline control data for the PBMC MGIA. These consolidate the lack of time-dependent effects on microbial restriction in BCG vaccinees to underscore the point that aerosolised BCG vaccination may not increase systemic immunity to Mtb. But the lack of ID vaccine comparator group is a critical gap in the data, meaning we can't actually interpret these findings.

The authors agree with my earlier critique that no measure of vaccine mediated 'protection' is shown. This means that no assessment of immune correlates of protection are possible. Therefore, the study offers no evidence that any of the immune responses described necessarily contribute to protective immunity.

I welcome the provision of data for absolute cell numbers retrieved from the BAL specimens, but these do not really help us interpret the percentage units provided in the BAL flow cytometry analysis. I am not sure why the authors' state that standardising to the epithelial cell count is not appropriate for controlling for technical variation in sampling.

I reiterate that it is misleading to present findings as being 'statistically significant' without multiple testing corrections. This should be addressed throughout the manuscript as previously stated.

The differential gene analysis at single cell level is also likely to generate substantial false positive results- at the very least, these should be acknowledged in the main body of the results narrative and the limitations in the discussion section. In addition, the authors should present per participant level data for selected DGEs of interest, so that we can assess the extent to which the findings are generalisable at least within the small sample. Otherwise, I have no confidence that the 'differentially expressed genes' are not driven by interindividual variation within groups.

I agree that validation of the findings of pathway analysis in different models is beyond the scope of the present study, but there is no reason why they could not use independently derived gene signatures for selected pathways to consolidate some of their findings in the existing single cell sequencing data.

NCOMMS-24-46679-T REVIEWER COMMENTS AND REBUTTAL

Reviewer #1 (Remarks to the Author):

In the current study the authors utilized a BCG human infection model to define initial innate and adaptive immune responses. I commend the authors on an excellent body of work. I have a few comments for the authors to consider:

1. It is unfortunate that the sample size was limited, which impacted the ability to draw robust conclusions about MAIT, iNKT, and polyfunctional T-cells. The authors do acknowledge this as a limitation. Future studies with larger sample sizes or optimized methods to obtain more BAL samples could potentially address these gaps. The authors would need to expand on this in the limitations section.

We thank the reviewer for their positive comments on our study. We have included the following two sentences in the discussion section.

We were limited in our flow cytometry analysis due to the numbers of BAL cells available. Some analyses were only carried out on a subset of volunteers who had more BAL cells retrieved. BAL cell count may be related to immune response and hence missing values may have confounded some of our results.

Further, this limited our ability to draw robust conclusions around certain cell types such as the Donor Unrestricted T cells and polyfunctional T cells. Future studies with a larger sample size or more focussed analysis could study these cells in more detail.

2. In figure 2 it is not clear how the authors phenotyped the APCs? I assume it is a combination of macrophages and DCs. Did the authors look at pDCs and cDCs? Can they draw any conclusions from this data. It does appear as if the cDC population increases in the BCG challenged versus placebo participants after 2 days post-challenge for the single cell sequencing data. The increase in the cDC population in response to BCG challenge is noteworthy and could suggest a differential role or response of cDCs compared to other APCs. The authors should clarify their methods for phenotyping these cells and discuss the implications of their findings in relation to the cDC increase.

Information for how cells were phenotyped can be found in the supplementary information (Extended data 8). "APCs" are a combination of dendritic cells, macrophages and monocytes. There were insufficient flow cytometry channels limiting colours as well as insufficient cells for meaningful read outs of subtypes of APCs.

3. It seems strange that the authors did not look at T-cell memory responses in the current study. The authors need to address this gap or justify why it was not included in their study.

All phenotyping is shown in the Appendix. These were BCG and *Mtb* naïve subjects and the aim of looking at these very early time points (D 2 and D7 post infection) was to characterise the innate immune response. The second manuscript, currently in preparation, analyses the groups with later BAL time points (D14, 28 and 56) and includes an analysis of memory T cells in the blood. BAL cells are always limiting and there are never enough for all analyses and conventional flow cytometry. Flow cytometry panels as well as exploratory analyses were different between this early phase of the

study (D2 and D7) focussing on innate markers, compared to the later timepoints (D14, 28 and 56) which focussed on adaptive markers.

Hence our focus here in this manuscript on early innate responses.

4. What was the activation status of the neutrophils in the BAL versus the blood pre- and post-challenge. Neutrophils in the BAL could show differential activation or functional responses compared to those in the peripheral blood, which could be relevant for understanding the local immune response in the lungs.

Neutrophils in our paper are already defined as CD16+ CD66b+ which represent activated neutrophils (immature neutrophils were too difficult to distinguish from other innate cells based on the markers in our panel). We were limited to the number of channels in the flow cytometer so were unable to further define the neutrophils. We agree this would be an interesting and important next step to investigate in subsequent studies.

5. No differences were observed for the MGIA assay with PBMCs post BCG inhalation versus baseline. Could this be explained by the differences in the memory immune responses in the lung versus the peripheral blood? Did the authors attempt to use BAL cells in these assays? Maybe a more robust assay would have been to use the mycobacterial stasis assay used by other groups by counting CFUs? Can the author's comment.

Thank you for raising this important point. We agree it would be very interesting to explore an MGIA using BAL cells. Unfortunately, BAL cells are always limited in number, and it was not possible to do this here as we used all the BAL cells for the innate flow cytometry panels. We do plan to establish a BAL MGIA in subsequent aerosol BCG studies. We mention these plans in the discussion (*"The inability to detect a change in control using the ex-vivo MGIA, despite clinical control of BCG infection, may be due to lack of sensitivity using PBMCs after an aerosol infection. Improved control has been shown on a murine lung MGIA.[27] Further work is in progress to optimise a BAL MGIA in humans."*)

6. HAIN sequencing was used to detect mycobacteria in the BAL. As the authors are aware the assay would not be able to discriminate between live and dead bacteria in the lung. A positive signal may simply have been generated by free genomic material in the lung. The authors need to comment on this in the manuscript.

We agree with the reviewer's assessment that mycobacteria detected using HAIN would detect both dead (intact and degraded) and live BCG. This is why we also reported on culture positive BCG using the Mycobacterial Growth Indicator Tube (MGIT) system and used culture positive BCG as the primary read-out (and note we saw a clear drop in detection from Day 2 to Day 7). HAIN sequencing was used simply for speciation to ensure the culture positive samples were BCG and not another incidental mycobacteria (i.e. NTM). To ensure clarity we have added the following to the results: *"BCG was detected by HAIN™ genotyping in the BAL from all twenty volunteers post BCG-infection, which could represent live, dead and/or fragmental BCG material."*

7. In the results section, line 199, the incorrect figure is referenced. "Fig 4D" should be "Fig 4E". We apologise for this error. Thank you for picking this up. This has now been amended.

Reviewer #2 (Remarks to the Author):

In their manuscript entitled “Early immune responses in the lung and blood following a randomised controlled human inhaled infection with aerosolised attenuated Mycobacterium Bovis BCG in healthy, BCG-naïve, UK adults” the authors further characterize the BCG controlled human infection model, describing in great detail both the systemic immune response and the respiratory response found in BAL fluid and biopsies. While the ability to follow the participants according to the original protocol was constrained by the COVID-19 pandemic, there is significant new information in this manuscript. As had been previously reported, the inhaled BCG at the dose given was well tolerated and immunogenic. The participants who received BCG had more significant symptoms than the controls who inhaled saline. The authors found that the immune response to inhaled BCG was faster than what had previously been reported for intradermal administration, or what was seen in natural mycobacterial infection. While there was not much of an immune signal after 2 days in either the blood or BAL, there was a significant immune response to the BCG by day 7, including an IFN gamma response. The paper is clear and easy to read.

We thank the reviewer for their positive comments on our study.

My comments/questions are below

1. In the previous study (Satti, Lancet 2024), the dose of the challenge was lower than anticipated due to both the variability of BCG in the vials- and also the loss of some of the dose in the nebulizer. In this manuscript, the authors state the BCG when reconstituted was 0.5 Log lower than stated on the label. How was this controlled for in the current study? Did the dose given take that lower level into account?

As this was a clinical trial, protocol defined dosing was stated according to what was on the vial. The dose of BCG was a “prepared or loaded dose” of 1×10^7 CFU based on our previous experience of tolerated infection dose (Satti, Marshall et al, Lancet Infectious Diseases 2024). The 0.5 log lower recovery from the BCG vial seen here is well within the known variability in BCG recovery within the starting vial (i.e. consistent with what we saw previously, as stated in the results). While it is difficult to quantify the losses due to nebulisation, as we followed the same methods as in our previous study (Satti, Marshall et al, Lancet Infectious Diseases 2024) we would not expect that the delivered dose would have varied significantly from that given in our previous trial (Satti, Marshall et al, Lancet Infectious Diseases 2024) i.e. estimated as 1×10^4 - 1.5×10^5 CFU delivered into the lung We therefore gave the expected dose and not a lower dose. We have clarified this in the discussion.

Edited wording: *“The “prepared dose” of 1×10^7 CFU was chosen based on our previous experience of tolerated “prepared or loaded” dose [18]. As reported in our previous study, the dose administered into the lungs was likely 1×10^4 - 1.5×10^5 CFU, 1.5-3 log lower than 1×10^7 CFU, due to the expected losses during nebulisation and the known variability in BCG within the starting vial [18, 22, 26].”*

2. Results, Enrolment, Line 81-82:

The authors state: “Fourteen follow up visits (D56, D84, D168; all Group 2 volunteers) were conducted as telephone visits”. This sentence is confusing. Were all of the Group 1 D56-168 visits remote? In addition to the Group 2 visits? But the Group 2 volunteers had their Day 7 BALs done?

We apologise for the wording of this sentence. To improve clarity, we have edited the sentence within the results section and added in an extended data table (new Extended data 1A) to provide more information on the altered visits.

“Due to the impact of the COVID-19 pandemic, blood or sputum was unable to be collected for some later follow up visits in Group 2 volunteers (from D56), in accordance with UK government policy (Extended Data 1).”

3. Only 60% of volunteers who were given inhaled BCG had viable organisms recovered on BAL on day 2 after challenge (and an even lower percentage on day 7). Could this be due to inadequate challenge dose not leading to an infectious “take” in those who had negative cultures?

We think this is primarily because the cells from the BAL were taken for use in the flow cytometer to characterise the innate immune response – the primary aim of the study. Subsequent work with different volunteers has shown that BCG recovery is higher when all BAL cells are used in the BACTEC MGIT.

We have added a sentence regarding this in the discussion to provide clarity:

“BAL cells were prioritised for immune analysis and hence only BAL fluid (BALF) was used to culture and detect BCG. The BCG recovery rates from bronchoscopy described in this study are likely an underestimate. Subsequent work has shown that BCG recovery is higher when all BAL cells are used in the BACTEC MGIT (Harris et al, unpublished data).”

4. Results section 4. Line 116.

It is unclear to this reviewer why the PBMC Mycobacterial Growth Inhibition assay did not demonstrate the ability of cells to control the BCG.

See response to reviewer 1 above. The effect may be limited to the pulmonary compartment and hence not detectable with sufficient sensitivity systemically. We plan to establish a BAL MGIA in subsequent aerosol BCG studies. We mention these plans in the discussion (*“The inability to detect a change in control using the ex-vivo MGIA, despite clinical control of BCG infection, may be due to lack of sensitivity using PBMCs after an aerosol infection. Improved control has been shown on a murine lung MGIA. Further work is in progress to optimise a BAL MGIA in humans.”*)

5. Figure 2: Eosinophils, and possibly neutrophils and NK cells

In looking at the figure, it seems that the eosinophils (and possibly neutrophils and NK cells?) are higher on Day 7 vs. Day 2 regardless of whether they were challenged with BCG or inhaled Saline. In fact, it is significantly higher in the saline control group than those challenged, albeit the numbers are small. Could it not be just a reaction to the inhalation rather than to the BCG challenge?

The only panel which seems to be convincingly different between the control and challenge on day 7 is the APCs.

Yes, we agree that the increase seen in Day 7 controls in these cells makes some of the data difficult to interpret. We discuss the limitations of the study especially due to small sample size of controls in the discussion (see text below). We think the increase in saline controls in these cells may be due to a technical error (all Day 7 control samples had bronchoscopies on the same day by chance), rather than a reaction to inhalation, however, given volunteers who inhale saline reports sore throat and cough at a similar rate to post-BCG infection (consistent with our earlier study (Satti, Marshall, et al, Lancet Infectious Diseases 2024), which may be another explanation for the results we present. We have therefore added the following into the discussion.

Wording in the discussion:

“We saw eosinophils increase in the BAL at D7 compared with D2, however the high eosinophil count in D7 saline controls make these findings difficult to interpret.”

“This study was designed as an initial exploratory study, not powered for statistical significance, and as such numbers, particularly in the saline control groups were low. For some markers, (such as Siglec-8 to detect eosinophils, CD56 for NK cells and IFN- γ) the D7 saline control samples on flow cytometry stained with higher frequency than D2 control samples (this was a trend only due to low sample size). All three D7 saline control BAL samples were processed, by chance randomisation, on the same day. Hence the difference in staining of some markers may have been due to a technical error. The results from D7 saline controls therefore need to be interpreted with caution.”

Addition: “Alternatively, the increase in eosinophils, neutrophils and NK cells in day 7 saline volunteers (and BCG-infected volunteers) could be due to a reaction to the inhalation itself. Consistent with findings from our previous study [18], volunteers who inhaled saline reported AEs such as cough and sore throat at a similar rate to BCG-infected volunteers.”

6. Figure 3: Panels G, H, I

What are the colors? There seems to be a missing legend.

The legend is to the left of panel I. We apologise if this was not clearly visible. We have pasted the figure below for your review.

Corresponding figure legend text: *“(G), CD161+ T cells (H) and iNKT cells (I) were able to be further defined by CD4 CD8 subsets; median % parent. DN: double negative (CD4-CD8-).”*

Minor comment:

Methods 2.2 Bronchoalveolar lavage and sputum processing

Line 524: What is the abbreviation BALF?

Bronchoalveolar lining fluid. Thank you, we have spelt this out the first time we use it in the revised manuscript.

Reviewer #3 (Remarks to the Author):

A descriptive study of immune changes in bronchoalveolar lavage (BAL) and systemically following aerosolised Bacille Calmette Guerin (BCG) challenge in immunologically naïve humans is presented. Follow up was disrupted by the COVID-19 epidemic.

Results of interest include that

1. Inhalation of what appears quite a substantial dose of bacteria (intended as 1×10^7) was in general well-tolerated
2. It was difficult however to recover live bacteria from subsequent BAL being found in 6/10 on day 2 and 2/10 on day 7.
3. Distinct changes in the cellular composition of BAL were induced by BCG mostly clearly of neutrophils, NK and DURT cells and both CD4 and CD8 positive lymphocytes. Such changes were not as prominent in peripheral blood.
4. By day 7 in BAL the frequency of CD4 and CD8 (and DN and DP) positive cells that were positive for Interferon (IFN)-gamma upon restimulation with BCG increased
5. A marked transient increase in the frequency of peripheral cells producing IFN-gamma on restimulation was observed at day 7
6. IgA and IgG antibody responses to BCG in both BAL and serum were not pronounced.
7. Single cell RNA sequencing of BAL from 6 donors (3 controls, 3 BCG) at varying timepoints confirmed BCG activation of both innate and adaptive immune pathways

The work appears well-performed and is technically challenging. Some of the results are intuitive and in line with what might have been expected although the ostensible induction of an acquired T cell response both in BAL and peripherally in naïve donors is, as the authors say, quite interesting and might not have been so predictable given results of studies in animals challenged with virulent strains of mycobacteria.

We thank the reviewer for their positive comments on our study.

The authors may consider the following

1. The study was conducted over 4 years ago. It is acknowledged the authors allude to disruption due to COVID-19 which is understandable but also mention intent to bronchoscope at later time points (D14, D28, D56). Surely these timepoints have been collected yet the intent to present data is described as being for a future paper

The results from this study were always planned as two separate papers given the volume of work and the different focus of research question. The focus of this paper is early innate response, while the focus of a later paper will be on the subsequent adaptive response. To that end, the flow panels were different between these early two groups (D2 and D7) and subsequent later groups (D14-56), as given BAL cell numbers were rate limiting, we could not comprehensively evaluate innate and

adaptive immune responses in every group but had to focus attention on early (innate) or later (adaptive). Data from these later groups (D14, 28 and 56) cannot be compared directly.

We agree that the sentence may be misleading as since the finalisation of this manuscript with subsequent editing and review, the analysis of the follow-on groups is close to being finalised. The sentence has therefore been amended to the following: *“Volunteers with bronchoscopies at later timepoints (D14, D28, D56) were also subsequently enrolled into this study. This clinical study was designed and planned from the beginning to analyse the innate and adaptive immune response induced in humans after a defined time point mycobacterial infection. BAL samples from the first two groups (with bronchoscopies at D2 and D7 post infection respectively) were analysed primarily for innate immunity. Samples from the subsequent 3 groups (Bronchoscopies at D14, D28 and D56 respectively) were analysed primarily for adaptive immunity. It was not possible to do both innate and adaptive analyses for all volunteers due to the limited cell recovery from the BAL fluid. Flow cytometry panels as well as exploratory analyses were therefore different between the early phase of the study (Groups 1 and 2, with bronchoscopies at D2 and D7 respectively) focussing on innate markers, compared to the later Groups 3-5 (with bronchoscopies at D14, 28 and 56 respectively) which focussed on adaptive markers. Data from Groups 3-5 (bronchoscopies at D14, 28 and 56) cannot be compared directly with Groups 1-2 and will be presented in a subsequent manuscript.”*

2. Flow cytometric results throughout are reported as percentage so it is inevitable some subsets will appear to fall if others rise. A supplementary table stating the actual numbers of cells recovered at BAL in each donor (or a median) would help put these percentages in perspective even if it is accepted absolute values would be difficult to calculate because of variable recovery and volume of BAL fluid.

We have added a table to Extended Data 8 showing the BAL cell counts for each of the groups.

3. The authors acknowledge that the inoculum did not turn out as high as expected, and also that the aerosol delivery will distribute this widely throughout the lung which in turn will undoubtedly affect recovery of bacilli at subsequent bronchoscopy. Nevertheless, widespread infection with a high dose of an avirulent Mycobacterium is counter to widespread belief that natural human infection by *M. tuberculosis* may arise from very few virulent bacilli that establish a localised lesion. The authors do discuss this but it is a limitation. Notwithstanding that the HAIN was positive, the rate of disappearance of viable bacilli is quite striking especially in ostensibly unsensitised donors (see below). Overall is what is set up in this model a transient very mild pneumonitis rather than a localised granulomatous focus?

The inoculum was as expected (i.e. consistent with our previous study – Satti, Marshall, et al, Lancet Infectious Diseases 2024). We have edited the manuscript to clarify this as we agree that the wording is misleading. Edited wording in the discussion: *“The “prepared dose” of 1×10^7 CFU was chosen based on our previous experience of tolerated “prepared or loaded” dose [18]. As reported in our previous study, the dose administered into the lungs was likely 1×10^4 - 1.5×10^5 CFU, 1.5-3 log lower than 1×10^7 CFU, due to the expected losses during nebulisation and the known variability in BCG within the starting vial [18, 22, 26]”.*

The aim in this study was to induce a transient and mild infection model at a tolerated dose of BCG. It would not be ethical to develop a disease model in humans where we induce granuloma. We

expect healthy immunocompetent people to clear BCG. Aerosol delivery best mimics the natural route of exposure to *Mtb*. It would not be ethical to administer virulent *Mtb* and we would not expect any disease manifestations after BCG delivery in immunocompetent people. We consider this model of aerosol BCG is valuable as it may provide novel insights into immune mechanisms that mimic the natural clearance of an early *M.tb* infection where there is also no establishment of a granuloma (potentially 90%+ of *M.tb* cases). One aim of this work would be to use subsequent validated immune data to inform the development of a vaccine or prophylactic therapeutic for prevention of infection (not prevention of disease).

The BCG BAL recovery in this study was limited as we used all the BAL cells for the flow cytometry analysis (which was the primary objective of the study, not quantifying BCG recovery). We have added a sentence regarding this in the discussion to provide clarity: *“BAL cells were prioritised for immune analysis and hence only BAL fluid (BALF) was used to culture and detect BCG. The BCG recovery rates from bronchoscopy described in this study are likely an underestimate. Subsequent work has shown that BCG recovery is higher when all BAL cells are used in the BACTEC MGIT (Harris et al, unpublished data).”*

4. The very substantial but transient peripheral response is also interesting. The authors discount that this may have arisen via prior sensitisation because of careful selection of donors. However the day 0 peripheral ELISpot response to BCG Danish is not entirely negligible especially bearing in mind the scale of figure 4E.

We agree completely with this comment. The D0 ELISpot response could suggest some prior mycobacterial sensitisation. This is why we say in the discussion that the kinetics of the BCG induced immune response require further investigation. Further work is needed to interrogate the baseline samples further to determine if there is any immunological evidence for prior sensitisation.

An aside is why was BCG Danish chosen to restimulate when the challenge was BCG Pasteur?

The challenge was with BCG Danish and the in vitro restimulation with BCG Pasteur, this was due to the difficulty in obtaining BCG Danish as we had just come out of a world-wide shortage of BCG Danish. We therefore wanted to reserve the BCG Danish for the clinical trial and therefore used the BCG Pasteur lab stocks for stimulation. As BCG Pasteur was used for all in vitro work for all volunteers, and there is no evidence that the two strains differ in protective efficacy, we do not think this detracts from our data.

5. In Figure 4D reported responses are heavily influenced by outliers

This study was designed as an initial exploratory hypothesis-generating study. We agree the data is limited due to small numbers, and inherent large inter-individual variability. This is why we have presented all data points, so readers are able to see and understand the raw data. It is also why we are careful to report that any detectable differences were lost after correction. However, this is the first time anyone has looked in the lung for immune responses so soon after an aerosol mycobacterial infection in humans and we consider the data worthy of publication.

6. The reason for missing data in Figure 3 panel 1 presumably relates to the number of cells available and should be stated in the legend

Yes that is correct. As stated under Methods 2.8 Flow cytometry detection of certain cell types was limited due to BAL cell numbers “BAL cells were allocated sequentially starting at Panel 1. The number of panels used for each volunteer was limited by the amount of BAL cells available.”

However, we agree we could be clearer about this, and we have added the detail as suggested in the Figure legend. “Bronchoalveolar lavage (BAL) post-BCG or saline inhalation or blood post-BCG inhalation stained with Panel 2 ($\gamma\delta$ T cells and CD3+CD56+ cells, whole blood) and Panel 3 (MAITs, iNKTs, CD161+ T cells, PBMCs) and analysed using an LSR Fortessa flow cytometer.

“As Panel 3 was third priority, there were only sufficient BAL cells for three D2 BCG-infection and four D7 BCG-infection BAL samples to be stained for detection of MAIT, iNKT and CD161+ T cells”

7. Given expansion of neutrophils why do these appear so infrequent in the sc RNA seq?

Granulocytes are difficult to capture on RNA-seq as they have low RNA content and high RNA inhibitory factors. Neutrophils are short lived and highly susceptible to degradation, particularly if placed on ice (part of the standard method for RNA processing). This, combined with the increased friability of BAL cells, likely explains why granulocytes were detected in low numbers in the scseq. A focussed study using conditions optimised for BAL granulocytes, such as using RNase inhibitors, minimising processing time or optimising temperature, would be required, if it was considered important to study neutrophils.

8. With respect to legends these excessively emphasise statistical testing and as such are dense and do not really explain what was done. It may be the journal style but I suggest stating the statistical tests on the last line of each legend and concentrating on a brief description of the data.

We would be happy to edit the legends as suggested pending editorial advice.

9. In the methods line 527-528 it is stated BAL cells were resuspended to achieve 1×10^7 lymphocytes/mL. Do the authors mean lymphocytes or all cells?

The text is correct as written. We standardised for lymphocyte concentration across PBMCs and BAL to ensure consistency with cell responses.

10. Spell out OP-RU and 6-FP in methods

Apologies – we have now done this (under 2.8 Flow Cytometry)

11. Line 796 how can the study have been funded in whole, or in part, by Wellcome Trust?

This is standardised wording that the Wellcome Trust has asked us to use for acknowledging grant funding. This clinical study was funded as part of HMcS Investigator Award.

12. Reference 44 is now published as J Exp Med. 2021 Oct 4;218(10):e20210469. It is not, as stated, a review. The lead author of reference 36 was Brill, KJ and Kaufmann, SHE was not an author at all.

Thank you. We have updated these references.

13. Other issues with referencing include lack of italics for species names (8,14,22,29,30,32-34,36,40,43,45,46-48,52); page numbers missing (2,3); year missing (18).

Thank you, we have updated these references.

Reviewer #4 (Remarks to the Author):

Summary

This manuscript aims to describe human in vivo local and systemic immune responses to aerosolised BCG vaccination. It is the first such study to do so. It evaluates cellular responses by flow cytometry and single cell RNA sequencing, some evaluation of antibody responses, and uses and an ex vivo peripheral blood mononuclear cell-based mycobacterial growth inhibition assay (MGIA) as surrogate measure of protective immunity. The opportunity for novel insights is driven predominantly by bronchoalveolar lavage sampling after aerosolised vaccination, but the data presented are limited to day 2 or day 7 BAL sampling time points aiming to focus on 'early' responses in the present report. For the most part, the interpretation of the data is critically limited by inadequate statistical power to make confident conclusions. The most robust finding is evidence for enrichment of T cells and antigen-specific Th1 responses by flow cytometry of BAL samples that is supported by evidence for increased TCR and IFN-signalling in single cell RNAseq data, along with increased frequency of circulating antigen specific T cell responses at day 7 identified by ELISpot. I don't find this a particularly significant advance, but the fact that it is being reported for the first time in this context has some incremental value, albeit preliminary in nature.

Major comments

There is no evidence of time-dependent increase in protective immunity generated by the aerosolised BCG vaccine in the MGIA data compared to pre-vaccination samples. Data from the control group receiving aerosolised saline are not presented- I think they should be included. However, it is not clear whether we should interpret this as evidence for lack of vaccine efficacy overall, lack of systemic protection in contrast to the potential of local protection in the respiratory tract, or simply that the MGIA is not a suitable outcome measure for protective immunity. I am aware that the authors have previously reported MGIA as evidence for protective immunity in vaccinees receiving conventional intradermal BCG, but is not included as a comparator group in the present study, so difficult to know what to make of the data.

As all volunteers controlled the BCG infection as evident clinically and by negative induced sputum sampling, we think this is an issue with a PBMC-based MGIA to detect an effect after aerosol BCG. We agree this could be because the effect may be limited to the pulmonary compartment and hence is not detectable using a PBMC-based MGIA. We plan to establish a BAL MGIA in subsequent aerosol BCG studies. We mention these plans in the discussion (*"The inability to detect a change in control using the ex-vivo MGIA, despite clinical control of BCG infection, may be due to lack of sensitivity using PBMCs after an aerosol infection. Improved control has been shown on a murine lung MGIA. Further work is in progress to optimise a BAL MGIA in humans."*).

As stated by the reviewer we didn't have an ID BCG control group in this study. This was due to the fact that the work was to assess the innate immune response after a defined time point human mycobacterial challenge. Hence the most important control group was inhaled saline.

We have updated the MGIA figure in extended data 4 to show the saline control group data.

The fact that no measure of vaccine mediated 'protection' is shown, means that no assessment of immune correlates of protection are possible. Hence the study cannot distinguish between immune responses that might be protective, inconsequential, or even harmful. The idea that any or all of the vaccine induced responses detected are likely to be contributing protective immunity is wholly speculative.

We agree with this comment. The aim of the study was for hypothesis generation with the requirement to validate any findings – we have reviewed the language throughout to emphasise this and the limitations of findings in the study. In addition, we have added the following sentence into the discussion:

“In addition, while this is a model of BCG control, not all immune responses described may be contributing to this control. Immune responses seen here may be contributing to protective immunity, but many may be inconsequential or detrimental to mycobacterial clearance.”

The statistical assessments of immune cell frequencies in BAL samples are muddled and misleading. First, the use of percentages to quantify cellular infiltration is uninterpretable- do these represent increases in absolute cell number or changes in proportions? The authors should really present some measure of changes in absolute number standardised by sample volume or another constant (eg epithelial cell frequency).

Absolute values were not possible to calculate (or would be meaningless) because of variable recovery and volume of BAL fluid. Epithelial cell frequency is also variable due to the inconsistency of bronchoscopy volume and hence would also not provide a constant. Presenting flow cytometry data as a percentage of parent is standard practice for cytokine responses such as IFN- γ as is done here. Other data is shown as % grandparent such as leucocytes or CD3- cells, which we consider appropriate. This shows the expansion of different cell populations (with corresponding reduction in others). We agree it doesn't reflect changes in total cell numbers but is a useful and standard measure for showing changes in the character of the total relevant cell population. We have included a table showing the median of absolute cell numbers from the BAL for each experimental group. As there were more cells retrieved at D2 compared to D7 any read-out of changes in absolute cell number subset would be misleading, emphasising the importance of expressing cell types as % of a relevant grandparent population.

Second, given the study design includes saline control recipients, the principal analysis should really focus on whether there are changes attributable to BCG compared to saline controls at the same time point. For example, the authors state “There was an increase in frequency of eosinophils, neutrophils, NK cells and the cytotoxic CD16+ NK cell subset in the airways of volunteers infected with aerosol BCG 123 at D7 compared with D2 or saline controls (Fig. 2)”. In fact, there is no evident statistically significant increase between D7 BCG and saline controls for of eosinophils, neutrophils or NK cells.

We consider a comparison between D2 and D7 post-BCG infection is a legitimate analysis and provides interesting and useful data between groups of sufficient size (up to 10 volunteers per

group) to meet our primary aim of generating hypotheses for immune markers for further investigation, and this was the basis on which ethical approval was given for the study. Given the small numbers of saline controls at each timepoint, comparison with saline controls and their corresponding BCG group alone would not yield data of sufficient value. Further, as discussed in the manuscript, by chance all Day 7 saline controls were collected and processed on the same day increasing the risk of technical error for these samples.

Due to the substantial number of results reported in this paper and the limitations on word count we have grouped results in logical subject order, such as in the example given above regarding innate cells. Each of these statements are followed with the statistical results making it clear which groups are statistically significant within each cell subset. However, we agree in the example given above, that as only CD16+*NK* cells differed between D7 BCG and saline and all other cell types differed between BCG D2 and D7, for clarity this should be split into two sentences which is what we have done

Third, reporting of statistically significant findings should only be reported after multiple testing corrections. If the authors limit their testing to comparisons with contemporary saline controls, then they may incur less multiple testing penalty than performing all pairwise testing. This critique applies to all flow cytometry panels in Figs 2, 3 & 4, the ELISpot data in Fig 4, and the PPD-specific antibody measurements in Fig 5.

We report all data as statistically significant results corrected for multiple comparisons. However, as this study was designed as an exploratory hypothesis generating, we believe limiting the reporting of results to only those that are significant after controlling for multiple comparisons would miss other potential signals which is why we have also reported significant results prior to correction.

We understand that an alternative to this, as suggested by the reviewer, could be to only report saline v BCG results and hence reduce the number of corrections required, however as explained above, given the small sample size for saline controls we feel this would miss a large number of potentially useful and interesting immune signals which we see when comparing D2 and D7 BCG infection.

It's a shame the authors chose to undertake T cell bead enrichment of the day 2 samples, which as they acknowledge, precluded any comparison of differences in cluster abundance between these time points. The UMAP figures presented in Fig 6A do not provide any particular analytical value. The module and pathway enrichment analysis of differential gene expression between aerosol BCG vaccinees and saline controls is helpful, and supports the finding of increased IFN γ and TCR signalling at day 7 following BCG. However, it's not clear whether the DGE analysis is at the level of individual donor (ie pseudobulked cluster data for each individual) or individual cell level. I suspect it is the latter, in which case we can't assess to what extent the findings are generalisable between BCG vaccine recipients, or the results are driven by one or a subset of individuals.

The analysis was indeed done on an individual cell level due to low sample size (only three volunteers in each group). This was a hypothesis generating study and findings will be validated with larger sample sizes in a follow on study. We have added some wording in Methods paragraph 2.9.5 DGE analysis to make it clear the analysis was at the cell level.

On a separate note, biological inferences made on the basis of pathway enrichment derived from DGE alone remain rather speculative – those that the authors wish to highlight as biological interesting really require orthogonal validation – for example by independent/experimentally derived gene expression signatures that represent the biological process or immune pathway of interest.

Yes we agree that this would be of value and are planning follow on studies. This is an experimental medicine hypothesis generating study – the importance is it is the first time a defined time point aerosol mycobacterial infection study was performed in humans looking at early, local immune responses. We agree the findings need validation in other models, but that is beyond scope of the work presented here.

Given the lack of statistical power and other weaknesses in the analyses presented, I find much of the points highlighted in the discussion overly speculative.

We have reviewed and modified the language throughout the discussion to address these concerns.

Minor points

The study participants are substantially enriched for young white females. This further undermines the lack of power to make generalisable observations.

We agree that future larger scale studies should include a more diverse population. We have added a sentence into the discussion to acknowledge this. *“This should include investigation in other populations to ensure generalisability of findings especially for those in M.tb endemic settings.”*

The authors should discuss the fact that BAL measurements are limited by sampling error. It’s a shame that they did not include at least two separate lobes/segments of lung to lavage so they can evaluate some measure of intra-individual consistency of these measurements. This is true not only for the microbiological measurements as well as the immunological assays. For example, how confident are they that yield of viable bacteria is not systematically different at different site in the lung?

We think that this is less likely as the BCG was delivered through aerosol inhalation which results in bilateral inhalation and deposition throughout the lung. We do acknowledge that deposition can vary in different parts of the lung. Due to the small sample size and expected large inter-volunteer variability in terms of immune responses, taking the BAL from a standardised area strengthens our study design as it removes another source of variability. Comparing intra-volunteer variability might be of interest for a future study, but we do not think that taking a sample from just one other site would have added more value to this initial study. Unless multiple lung sites are able to be sampled, from all lobes (which is difficult or almost impossible to do consistently in human subjects) this would not yield much useful data.

Regarding the yield of viable bacteria, this was not one of the primary or secondary objectives of the study. Cells from the BALF were taken for use in the flow cytometer to characterise the innate immune response – the primary aim of the study. And hence the actual BCG load at D2 or D7 may well be different to what is reported in the manuscript.

We have added a sentence regarding this in the discussion to provide clarity: *“BAL cells were prioritised for immune analysis and hence only BAL fluid (BALF) was used to culture and detect BCG. The BCG recovery rates from bronchoscopy described in this study are likely an underestimate. Subsequent work has shown that BCG recovery is higher when all BAL cells are used in the BACTEC MGIT (Harris et al, unpublished data).”*

In addition, we do report on induced sputum yield which samples from all parts of the lung.

The reference to BCG-specific monocytes (line 194) is particularly odd/misleading concept as there is no mechanism by which monocytes can be ‘specific’ for BCG. I assume they mean BCG-reactive monocytes.

Thank you for this pick up, we have changed the wording in the text.

REVIEWER COMMENTS

Reviewer #1 (Remarks to the Author):

I commend the authors on an excellent study. The authors acknowledged their limitations when using BAL cells. I am satisfied with the authors responses and the amendments to the manuscript.

We thank Reviewer 1 for these comments.

Reviewer #2 (Remarks to the Author):

The authors have satisfactorily addressed my questions in the submitted revised manuscript. I have no additional questions, except that one comment from another reviewer re: italicizing genus/species names in the references does not seem to have been done.

Thank you. We have corrected this.

Reviewer #3 (Remarks to the Author):

The authors have made a conscientious effort to address my comments and have introduced changes in the manuscript

Thank you.

Reviewer #4 (Remarks to the Author):

I reviewed the revised manuscript focussing on the authors' response to my previous comments. In general, it seems that the authors acknowledge most of the critique, but have not been able to address any of the fundamental weaknesses in their study. The manuscript offers some preliminary descriptive analysis of early immune responses in BAL samples following aerosolised BCG vaccination, but fails to offer any new biological insights.

I welcome the addition of saline control data for the PBMC MGIA. These consolidate the lack of time-dependent effects on microbial restriction in BCG vaccinees to underscore the point that aerosolised BCG vaccination may not increase systemic immunity to Mtb. But the lack of ID vaccine comparator group is a critical gap in the data, meaning we can't actually interpret these findings.

The authors agree with my earlier critique that no measure of vaccine mediated 'protection' is shown. This means that no assessment of immune correlates of protection are possible. Therefore, the study offers no evidence that any of the immune responses described necessarily contribute to protective immunity.

I welcome the provision of data for absolute cell numbers retrieved from the BAL specimens, but these do not really help us interpret the percentage units provided in the BAL flow cytometry analysis. I am not sure why the authors' state that standardising to the epithelial cell count is not appropriate for controlling for technical variation in sampling.

I reiterate that it is misleading to present findings as being 'statistically significant' without multiple testing corrections. This should be addressed throughout the manuscript as previously stated.

The differential gene analysis at single cell level is also likely to generate substantial false positive results- at the very least, these should be acknowledged in the main body of the results narrative and the limitations in the discussion section. In addition, the authors should present per participant level data for selected DGEs of interest, so that we can assess the extent to which the findings are generalisable at least within the small sample. Otherwise, I have no confidence that the

'differentially expressed genes' are not driven by interindividual variation within groups.

I agree that validation of the findings of pathway analysis in different models is beyond the scope of the present study, but there is no reason why they could not use independently derived gene signatures for selected pathways to consolidate some of their findings in the existing single cell sequencing data.

Comments from arbitrating reviewer on Reviewer #4's most recent concerns:

I reviewed the revised manuscript focussing on the authors' response to my previous comments. In general, it seems that the authors acknowledge most of the critique, but have not been able to address any of the fundamental weaknesses in their study. The manuscript offers some preliminary descriptive analysis of early immune responses in BAL samples following aerosolised BCG vaccination, but fails to offer any new biological insights.

I welcome the addition of saline control data for the PBMC MGIA. These consolidate the lack of time-dependent effects on microbial restriction in BCG vaccinees to underscore the point that aerosolised BCG vaccination may not increase systemic immunity to Mtb. But the lack of ID vaccine comparator group is a critical gap in the data, meaning we can't actually interpret these findings.

RESPONSE: The reviewer is correct and a major concern that I raised during review i.e. the authors observed no differences in mycobacterial stasis assay (MGIA) post-BCG nebulisation in peripheral blood effector cells. Seeing that no differences were observed it would have been critical to see if the authors would have observed the same effect with intradermal BCG. The authors also did not include MGIA with BAL cells. If a reduction in bacterial load was observed with BAL cells post-nebulisation, then the authors could have argued that the antimicrobial effect was compartment-specific (i.e. lung and not peripheral blood). The MGIA remains the best method to measure vaccine efficacy in vitro. The reason for this is that the MGIA is a much more powerful, direct, and biologically meaningful outcome measure than inference from biomarker-specific proxy markers of presumed protection (cytokines, etc) that are often used in human studies. Therefore the lack of M.tb stasis does not provide experimental evidence of in vitro efficacy post BCG nebulisation.

We thank Reviewer 4 and the arbitrating reviewer for their comments and detailed feedback.

We cannot directly compare the data presented with a group of subjects that received BCG by the ID route, as there was no such group in our study. The point of this study was not to compare aerosol vs ID BCG but to map the immune cell influx after aerosol BCG and compare this with saline. However, in previously published studies we have shown a reduction in BCG growth 4- and 8-weeks post-ID BCG compared to baseline in the MGIA; this is noted in lines 281-282 of the "tracked" manuscript. We acknowledge that including a BAL MGIA would have offered additional information, however, this was not feasible due to the limited availability of BAL cells for the assays already conducted. Currently, there is no BAL MGIA assay that has been developed and previous attempts to develop such an assay have not been successful (doi: 10.3389/fimmu.2018.01708). Our findings provide novel information on the early immune response following a defined time point aerosolised mycobacterial infection in humans.

The authors agree with my earlier critique that no measure of vaccine mediated 'protection' is shown. This means that no assessment of immune correlates of protection are possible. Therefore, the study offers no evidence that any of the immune responses described necessarily contribute to protective immunity.

RESPONSE: This was one of my major critiques of the manuscript. The reviewer is correct, while we have no exact proxy to measure protection, further evidence to support this hypothesis would have been more convincing. Data to further support the presumed protective phenotype post-BCG nebulisation could have additionally been provided by tissue resident T-cells, polyfunctional T-cell, MAIT and iNK data. It was also concerning that no differences in lung mucosal antibody (IgA) or IgG responses were observed post-vaccination. Wholistically together all this data would provide evidence to suggest a presumed protective phenotype post-BCG nebulisation.

Our aim in this study was to characterise and define early immune responses following BCG vaccination, rather than to establish correlates of protection. We acknowledge that knowledge about protective immunity can not be directly concluded from our findings. However, understanding these early immune responses is crucial for identifying potential biomarkers that could later be validated in other studies that could demonstrate an association with protective immunity.

I welcome the provision of data for absolute cell numbers retrieved from the BAL specimens, but these do not really help us interpret the percentage units provided in the BAL flow cytometry analysis. I am not sure why the authors' state that standardising to the epithelial cell count is not appropriate for controlling for technical variation in sampling.

RESPONSE: I disagree with the reviewer. Variation in frequency of cell counts do not affect the absolute percentage of expression by flow cytometry. If the authors reported their data as mean fluorescent intensity, then yes cell counts would need to be accounted for across the data.

I reiterate that it is misleading to present findings as being 'statistically significant' without multiple testing corrections. This should be addressed throughout the manuscript as previously stated.

RESPONSE: I think the reviewer is being far too critical in this instance. The participant group included in this study was well defined and controlled. Therefore, confounding variables which would need to be controlled in the analysis is redundant.

In the manuscript, we have stated where statistical significance was lost after applying correction for multiple comparisons. We believe this ensures transparency and prevents any misleading interpretations of the data.

Line 119 "tracked changes version": "There was an increase in frequency of eosinophils, neutrophils and NK cells in the airways of volunteers infected with aerosol BCG at D7 compared with D2 (Fig. 2), though the difference in neutrophils lost significance after correction for multiple comparisons. (Eosinophils: median D2BCG 0.4% (IQR 0.2;0.6) v D7BCG 1.1% (0.8;1.6), $p=0.01$; NK cells: D2BCG 0.98% (0.6;1.5) v D7BCG 3.5% (2.6;5.4), $p=0.0009$, Mann-Whitney with Dunn's correction)."

Line 151 "tracked changes version": "In contrast, the frequency of iNKT cells and $\gamma\delta$ T cells fell in the blood post-BCG infection, but significance was lost after correction for multiple comparisons (Fig 3)"

The differential gene analysis at single cell level is also likely to generate substantial false positive results- at the very least, these should be acknowledged in the main body of the results narrative and the limitations in the discussion section. In addition, the authors should present per participant level data for selected DGEs of interest, so that we can assess the extent to which the findings are generalisable at least within the small sample. Otherwise, I have no confidence that the 'differentially expressed genes' are not driven by interindividual variation within groups.

I agree that validation of the findings of pathway analysis in different models is beyond the scope of the present study, but there is no reason why they could not use independently derived gene signatures for selected pathways to consolidate some of their findings in the existing single cell sequencing data.

RESPONSE: This is possible and would strengthen the data. I am not sure why the authors ignored this request.

We thank the two reviewers for these comments. To be consistent with the analysis of the combined scRNA-seq data from Groups 1-5 in a subsequent publication, which is currently in preparation, we have now re-analysed the scRNA-seq data and performed the differential expression analysis at the individual level using the pseudo-bulk expression data. We have updated the methods of scRNA-seq data analysis in the Methods Section 2.9.1 to 2.9.5 accordingly. We have also updated the results of scRNA-seq data analysis in the results and discussion section.

To validate the findings of the pathway analysis, we selected an independently derived T1-T17 gene signature from a NHP study, whose expression in *Mtb* granuloma was correlated with the control of *Mtb* growth, and a PPD-response gene signature derived from *in vitro* PPD-stimulated BAL cells from another NHP study. We then calculated the score of these gene signatures in T and NK cell populations in different volunteers. The method of gene signature score is as follows (also see the Methods section 2.9.5):

Gene signature scores were calculated for each cell type and sample using the pseudo-bulk expression matrix. The pseudo-bulk expression matrix was normalised and log₂-transformed using the estimateSizeFactors and counts functions in DESeq2 (v1.34.0). The gene signature score for each cell type and sample was calculated as the average expression level of the genes within the gene signature.

We have added the following findings in the results section:

Previous studies have identified several “T1-T17” populations in Mtb granuloma whose abundance was negatively associated with bacterial burden. To investigate whether a similar transcriptional profile was induced at early time points following aerosol BCG challenge, we scored T cell and NK cell populations using the gene signature defining the T1-T17 subpopulation 1, characterised by markers of activation and motility, and T1-T17 subpopulation 2, characterised by markers of cytotoxic effector molecules. The score derived from T1-T17 subpopulation 1 showed increased expression in CD4+ T cells, MAIT cells, Tgd cells and NK cells at either D2 or D7. The score derived from subpopulation 2 predominantly showed increased expression in CD4+ T cells, MAIT cells, and NK cells (Figure 6C). Additionally, scoring T cell and NK cell populations using a PPD-response signature derived from in vitro PPD stimulation of BAL cells revealed increased expression in MAIT and NK cells on D2, with a trend towards increased expression in CD4+ T cells on D7.

We have acknowledged the limitation of the small sample size in the scRNA-seq data in the discussion section:

Our scRNA-seq analysis was also limited by the small number of samples, with only three samples from the BCG group and three from the saline group sequenced at each time point. Future studies incorporating larger sample sizes to improve the statistical power and robustness of the scRNA-seq analysis were required to confirm the results.